# Pre-existing immunity and vaccine history determine hemagglutinin-specific CD4 T cell and IgG response following seasonal influenza vaccination

Katharina Wild[1,2,3], Maike Smits[1,2,4], Saskia Killmer[1,2], Shirin Strohmeier[5], Christoph Neumann-Haefelin [1,2], Bertram Bengsch [1,2,6], Florian Krammer [5], Martin Schwemmle [7], Maike Hofmann [1,2], Robert Thimme [1,2], Katharina Zoldan[1,2] & Tobias Boettler [1,2 ✉]

Effectiveness of seasonal influenza vaccination varies between individuals and might be affected by vaccination history among other factors. Here we show, by monitoring frequencies of CD4 T cells specific to the conserved hemagglutinin epitope $HA_{118-132}$ and titres of IgG against the corresponding recombinant hemagglutinin protein, that antigen-specific CD4 T cell and antibody responses are closely linked to pre-existing immunity and vaccine history. Upon immunization, a strong early reaction is observed in all vaccine naïve participants and also in vaccine experienced individuals who have not received the respective seasonal vaccine in the previous year. This response is characterized by $HA_{118-132}$ specific CD4 T cells with a follicular helper T cell phenotype and by ascending titers of hemagglutinin-specific antibodies from baseline to day 28 following vaccination. This trend was observed in only a proportion of those participants who received the seasonal vaccine the year preceding the study. Regardless of history, levels of pre-existing antibodies and CD127 expression on CD4 T cells at baseline were the strongest predictors of robust early response. Thus, both pre-existing immunity and vaccine history contribute to the response to seasonal influenza vaccines.

[1] Department of Medicine II, Medical Center – University of Freiburg, Freiburg, Germany. [2] Faculty of Medicine, University of Freiburg, Freiburg, Germany. [3] Faculty of Pharmacy, University of Freiburg, Freiburg, Germany. [4] Faculty of Biology, University of Freiburg, Freiburg, Germany. [5] Department of Microbiology, Icahn School of Medicine at Mount Sinai, New York, NY, USA. [6] Signalling Research Centres BIOSS and CIBSS, University of Freiburg, Freiburg, Germany. [7] Institute of Virology, Freiburg University Medical Center, Faculty of Medicine,  University of Freiburg, Freiburg, Germany. ✉email: tobias.boettler@uniklinik-freiburg.de

Pandemic-prone viruses like influenza pose a permanent global health threat as antigenic shift and drift results in viral variants circulating around the globe. Seasonal vaccinations account for antigenic changes of the most abundant influenza strains and represent an important measure for prevention[1]. An effective immune response to the vaccine leads to the generation of neutralizing antibodies that are able to prevent viral entry[2,3]. However, great variations of the antibody titers elicited by the vaccine have been observed[4–6] with particularly poor antibody induction and impaired affinity maturation occurring in individuals receiving repeat vaccinations[4,7–12]. Maturation of antibody responses is a T cell-dependent process and recent data strongly suggest that this is also the case in the context of recall responses[13,14]. Thus, defining the role of influenza-specific T cells in the different outcomes after repeat vaccinations is of major relevance. Among CD4 T helper cells, T follicular helper (Tfh) cells are key players in the generation of antibody responses as they provide the fundamental signals to B cells in the germinal center (GC) reaction underlying affinity maturation. Tfh cells are characterized by expression of the CXC motif chemokine receptor 5 (CXCR5) and programmed death protein 1 (PD-1)[15]. Since GC-derived Tfh cells are difficult to assess in humans, circulating Tfh (cTfh) cells are commonly used as surrogates as they share phenotypic, functional, and clonal characteristics with lymphoid tissue-derived GC-Tfh cells[16–21]. However, the definitions used to identify cTfh cells are heterogeneous[22] and are mostly limited to the expression of CXCR5 (+/−PD-1) as well as CD38 and inducible T cell costimulatory (ICOS) to identify activated cTfh populations. The master transcription factor of Tfh cells, Bcl6[15] is not expressed on cTfh cells[18,23] and the precise expression patterns of other transcription factors that have been associated with Tfh development, such as T cell factor 1 (TCF-1) and TOX, remain to be identified[24,25]. Furthermore, they have not been established as lineage-defining markers for antigen-specific cTfh cells in humans as they are not exclusively active in the Tfh program[25,26]. Previous studies analyzing bulk cTfh cells after influenza vaccination identified a peak of activation on day 7 characterized by CD38, ICOS, and CXCR3 expression[27–30]. Similar to observations on antibody titers after repeat vaccination, this was more strongly pronounced in individuals without self-reported influenza vaccination in the preceding years[11]. In other studies, activation-induced marker or cytokine expression after antigen stimulation confirmed the presence of antigen-specific CD4 T cells within this population[23,31] and an increase of this population correlated with the amount[27,29,31,32] and the avidity[28] of the flu-specific antibodies, suggesting a mechanistic connection between cTfh responses and vaccine-elicited antibodies[10,11,30]. However, few studies have aligned the major histocompatibility complex (MHC) class II epitope sequences with the current vaccine strain or have phenotypically analyzed influenza-specific CD4 T cell responses at baseline and earlier than day 7 after vaccination[27,33].

Here we show by analyzing CD4 T cells specific for a highly conserved hemagglutinin (HA)-derived epitope of influenza A virus using tetramer technology that allows comprehensive and longitudinal characterization of circulating influenza-specific CD4 T cells that response patterns are closely associated with vaccination history. In addition, activation patterns of vaccine-induced, influenza-specific CD4 T cell responses correlate with vaccine-elicited antibody titers and baseline factors that might be able to predict the cellular and humoral response to the vaccine are observed.

## Results

### Expansion of influenza-specific CD4 T cells after vaccination.
Seasonal influenza vaccines are adapted to include the sequence mutations of the influenza virus strains predicted to be most common. In order to longitudinally analyze circulating influenza-specific CD4 T cells responding to the vaccine (Fig. 1a), we focused on two HLA-DRB1*01:01–MHC class-II-restricted HA3-derived epitopes (HA$_{118-132}$ VPDYASLRSLVASSG and HA$_{306-318}$ PKYVKQNTLKLAT) in 12 individuals (Table 1). While the latter (HA$_{306-318}$) has been employed in studies to analyze vaccine-elicited CD4 T cell responses[23,34], it has not been included in the annual vaccine strains between 2016 and 2020, harboring sequence variations on two to three positions in recent years (Fig. 1b and Supplementary Table 1). In contrast, HA$_{118-132}$ was conserved in all annual vaccine strains in that time span. Using pMHCII-tetramer technology, we analyzed HA$_{118-132}$- and HA$_{306-318}$-specific CD4 T cells ex vivo in HLA-DRB1*01:01-positive individuals who received influenza vaccination in 2016/2017, 2018/2019, 2019/2020, and/or 2020/2021 (see also Table 2 and Supplementary Table 1). We could identify an increase in HA$_{118-132}$-specific CD4 T cell frequencies as early as 4 days after vaccination (with few individuals already showing an increase on day 2), suggesting a direct effect of the vaccine on the expansion of this T cell subset (Fig. 1c, d and Supplementary Figs. 1 and 3a–c). In contrast, frequencies of HA$_{306-318}$-specific CD4 T cells did not change, indicating that the sequence variations in the vaccine strain prevented the activation of this T cell specificity. (Fig. 1c, d and Supplementary Fig. 2). The dynamics of HA$_{118-132}$-specific CD4 T cell frequencies showed strong heterogeneity between individuals. Therefore, we separated our cohort of vaccinees into three different groups according to their self-reported vaccination history: Blue = vaccinated in the previous year, turquoise = last vaccine received >1 year ago [range 2–10 years], red = vaccine-naive (Fig. 1e). As shown in Fig. 1f, vaccine-naive donors (red) and those with repeat vaccination within 2–10 years (turquoise) showed higher fold changes (FC) of HA$_{118-132}$-specific CD4 T cell frequencies compared to vaccinees with repeat vaccination within 1 year (blue), with some individuals almost maintaining baseline frequencies (e.g., #2-2019 and #5-2016). In addition, frequencies in vaccine-naive donors peaked earlier (around day 4) compared to the vaccine-experienced groups. Collectively, these data demonstrate that the expansion dynamics of influenza-specific CD4 T cells are heterogeneous among vaccinees but appear to be linked to the individual's vaccination history.

### Emergence of activation markers on HA$_{118-132}$-specific CD4 T cells.
Next, we assessed the phenotypic changes of HA$_{118-132}$-specific CD4 T cells after vaccination with regards to markers indicating memory (interleukin (IL)-7 receptor alpha, CD127)[35], proliferation (Ki67), and activation (CD38 and ICOS). These phenotypic analyses revealed a memory phenotype on HA$_{118-132}$-specific CD4 T cells at baseline, characterized by high expression levels of CD127 in the absence of activation or proliferation. In line with our observations on HA$_{118-132}$-specific CD4 T cell frequencies, Ki67 peaked early after vaccination, which was associated with a decrease of CD127 expression (Fig. 2a, b). Both changes were most prominently observed in vaccine-naive donors (red) and to a lesser extent in those with repeat vaccination within 2–10 years (turquoise). Similarly, co-expression of ICOS and CD38 on HA$_{118-132}$-specific CD4 T cells as early as day 4 post vaccination was significantly higher in individuals receiving the first seasonal vaccine (red), compared to those with repeat vaccination within 2–10 years (turquoise) and 1 year (blue, Fig. 2c). ICOS and CD38 co-expression on HA$_{118-132}$-specific CD4 T cells (on day 4) positively correlated with the FC of HA$_{118-132}$-specific CD4 T cell frequencies (from baseline to day 4, Fig. 2d), which was largely driven by the vaccine-naive donors. Among the recently vaccinated cohort (blue), the donors that had little to no

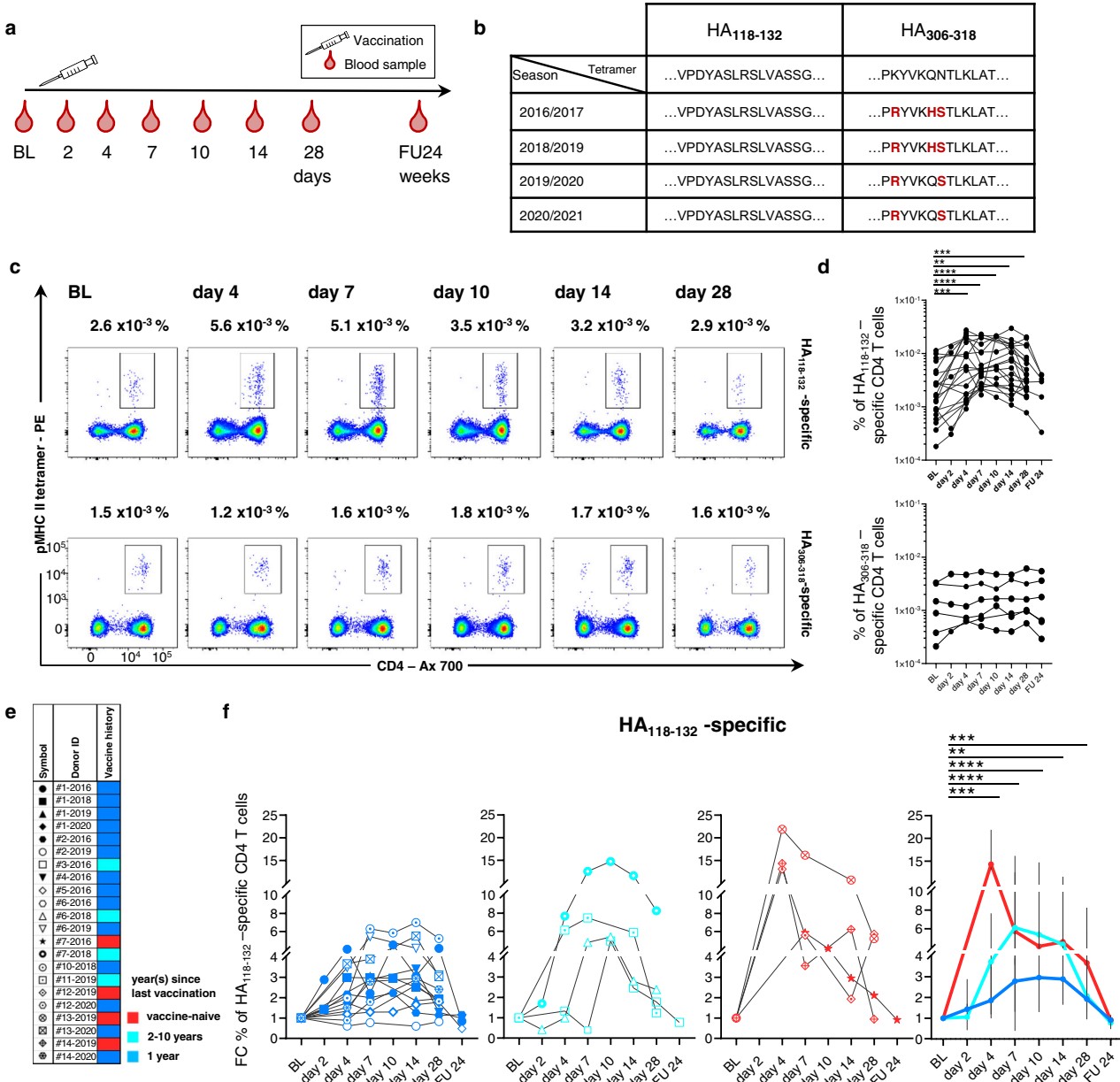

**Fig. 1 Recognition of the HA$_{118-132}$ epitope leads to expansion of pre-existing memory HA$_{118-132}$-specific CD4 T cells following seasonal vaccination.**
**a** Design of the vaccination study. **b** Epitope amino acid sequences of the tetramers and within seasonal vaccines. Changes of amino acids in comparison to the tetramer sequence are marked in red. **c** Representative pseudocolor plots of HA$_{118-132}$- and HA$_{306-318}$-specific CD4 T cells before/after vaccination. **d** Frequency of HA$_{118-132}$- and HA$_{306-318}$-specific CD4 T cells at the indicated time points (HA$_{118-132}$-specific for BL $n = 21$; day 2 $n = 5$; day 4 $n = 19$; day 7 $n = 20$; day 10 $n = 9$; day 14 $n = 21$; day 28 $n = 21$; FU24 $n = 6$; HA$_{306-318}$-specific for BL $n = 6$; day 2 $n = 3$; day 4 $n = 5$; day 7 $n = 6$; day 10 $n = 5$; day 14 $n = 6$; day 28 $n = 6$; FU24 $n = 6$). **e** Legend for donor symbols and colors indicating the vaccination history (red = vaccine-naive (4), turquoise = 2–10 years (4) and blue = 1 year (14) since last vaccination). **f** FC of HA$_{118-132}$-specific CD4 T cells at the indicated time points relative to BL. The responses are subgrouped by the vaccination history. (blue: $n = 13$, turquoise: $n = 4$, red: $n = 4$). The summary plot (right) shows the median with 95% confidence interval within the three subgroups. One symbol represents one response at the indicated time point. Two-tailed Wilcoxon matched-pairs signed rank test with Bonferroni correction was used for multiple between-group comparisons. All time points were compared to BL. In **d**, **f**, a $p$ value of 0.007 was considered statistically significant. **\*\***$p < 0.007$, **\*\*\***$p < 0.001$, **\*\*\*\***$p < 0.0001$. BL baseline, FU follow-up, FU24 24 weeks after vaccination, FC fold change.

change in HA$_{118-132}$-specific CD4 T cell frequencies (e.g., #2-2019 and #5-2016, Fig. 1f) nevertheless upregulated ICOS and CD38, demonstrating that all donors mounted an HA$_{118-132}$-specific CD4 T cell response by proliferation, activation, or both. We subsequently analyzed HA$_{118-132}$-specific CD4 T cell responses in more detail, including markers that have previously been associated with activation in response to vaccination (CXCR3) and

Tfh cell differentiation (CXCR5, PD-1). Results are shown as normalized mean fluorescence intensity and FC of HA$_{118-132}$-specific CD4 T cells frequencies at baseline on heatmaps (Fig. 2e and Supplementary Fig. 3d, e). These data display marked, although heterogeneous, alterations from baseline to days 4 and 7 after vaccination for all plotted parameters, appearing to separate the entire cohort into a group with activation signals dominating

**Table 1 Study participants and analyses of different vaccination responses.**

| Donor ID | Ethnicity | Years since last vaccination | Age | Sex | HLA-class II of interest | Season | Early response | Late response | Tetramer staining (HA306-318) | Tetramer staining (HA118-132) Panel 1 | Tetramer staining (HA118-132) Panel 2 | IgG ELISA |
|---|---|---|---|---|---|---|---|---|---|---|---|---|
| #1 - 2016 | White | 1 | 36 | M | DRB1*01:01 | 2016/2017 | | x | x | x | | x |
| #1 - 2018 | White | 1 | 38 | M | DRB1*01:01 | 2018/2019 | | x | | x | | x |
| #1 - 2019 | White | 1 | 39 | M | DRB1*01:01 | 2019/2020 | | x | | x | x | x |
| #1 - 2020 | White | 1 | 40 | M | DRB1*01:01 | 2020/2021 | | x | | | x | x |
| #2 - 2016 | White | 1 | 27 | F | DRB1*01:01 | 2016/2017 | x | x | x | x | | x |
| #2 - 2019 | White | 1 | 29 | F | DRB1*01:01 | 2019/2020 | x | | | | | x |
| #3 - 2016 | White | 2 | 46 | M | DRB1*01:01 | 2016/2017 | | | | | x | x |
| #4 - 2016 | White | 1 | 26 | M | DRB1*01:01 | 2016/2017 | No data day 4 | No data day 4 | x | x | | x |
| #4 - 2018 | White | 1 | 27 | F | DRB1*01:01 | 2018/2019 | | | x | x | | x |
| #5 - 2016 | White | 1 | 37 | M | DRB1*01:01 | 2016/2017 | | x | x | x | | x |
| #6 - 2016 | White | 2 | 39 | M | DRB1*01:01 | 2016/2017 | | | x | | | x |
| #6 - 2018 | White | 1 | 40 | M | DRB1*01:01 | 2018/2019 | x | | | x | x | x |
| #6 - 2019 | White | 0 | 23 | M | DRB1*01:01 | 2019/2020 | x | | | | | x |
| #7 - 2016 | White | 2 | 25 | M | DRB1*01:01 | 2016/2017 | No data day 4 | No data day 4 | x | x | | x |
| #7 - 2018 | White | 1 | 26 | M | DRB1*01:01 | 2018/2019 | x | | | x | | x |
| #7 - 2019 | White | 1 | 27 | M | DRB1*01:01 | 2019/2020 | | | | | | x |
| #7 - 2020 | White | 0 | 27 | M | DRB1*01:01 | 2020/2021 | | | | | | x |
| #8 - 2016 | White | 0 | 32 | F | Other | 2016/2017 | | | | | | x |
| #9 - 2016 | White | 2 | 33 | F | Other | 2016/2017 | | | | | | x |
| #9 - 2019 | White | 1 | 45 | F | Other | 2019/2020 | | | | | | x |
| #10 - 2018 | White | 1 | 46 | M | DRB1*01:01 | 2018/2019 | | x | | x | | x |
| #10 - 2019 | White | 10 | 47 | M | DRB1*01:01 | 2019/2020 | | | | | | x |
| #10 - 2020 | White | 0 | 40 | M | DRB1*01:01 | 2020/2021 | | | | | | x |
| #11 - 2019 | White | 1 | 28 | F | DRB1*01:01 | 2019/2020 | x | | | | x | x |
| #12 - 2019 | White | 0 | 29 | F | DRB1*01:01 | 2019/2020 | x | | | | x | x |
| #12 - 2020 | White | 1 | 29 | F | DRB1*01:01 | 2020/2021 | | x | | | x | x |
| #13 - 2019 | White | 0 | 30 | F | DRB1*01:01 | 2019/2020 | x | | | | x | x |
| #13 - 2020 | White | 1 | 24 | F | DRB1*01:01 | 2020/2021 | | x | | | x | x |
| #14 - 2019 | White | 1 | 24 | F | DRB1*01:01 | 2019/2020 | x | | | | x | x |
| #14 - 2020 | White | 0 | 25 | F | DRB1*01:01 | 2020/2021 | | x | | | x | x |
| #15 - 2019 | White | 1 | 27 | F | DRB1*01:01 | 2019/2020 | | | | | | x |
| #16 - 2019 | White | 0 | 24 | F | Other | 2019/2020 | | | | | | x |
| #17 - 2019 | White | 2 | 30 | M | Other | 2019/2020 | | | | | | x |
| #18 - 2019 | White | 10 | 21 | M | Other | 2019/2020 | | | | | | x |
| #19 - 2019 | White | 1 | 28 | F | Other | 2019/2020 | | | | | | x |
| #20 - 2019 | White | 2 | 26 | F | Other | 2019/2020 | | | | | | x |
| #21 - 2019 | White | 2 | 29 | F | Other | 2019/2020 | | | | | | x |
| #22 - 2019 | White | 1 | 25 | F | Other | 2019/2020 | | | | | | x |

**Table 2 Seasonal vaccines 2016–2020.**

| Influenza vaccination | Influenza Virus A (H3N2) | Supplier |
|---|---|---|
| Influsplit Tetra 2016/2017 | A/Hong Kong/4801/2014 (H3N2) | GlaxoSmithKline GmbH & Co. KG |
| Vaxigrip Tetra 2018/2019 | A/Singapore/INFIMH-16-0019/2016 (H3N2) | Sanofi-Aventis GmbH |
| Influsplit Tetra 2019/2020 | A/Kansas/14/2017 (H3N2) | GlaxoSmithKline GmbH & Co. KG |
| Influvac Tetra 2020/2021 | A/Hong Kong/2671/2019 (H3N2) | Mylan Healthcare GmbH |

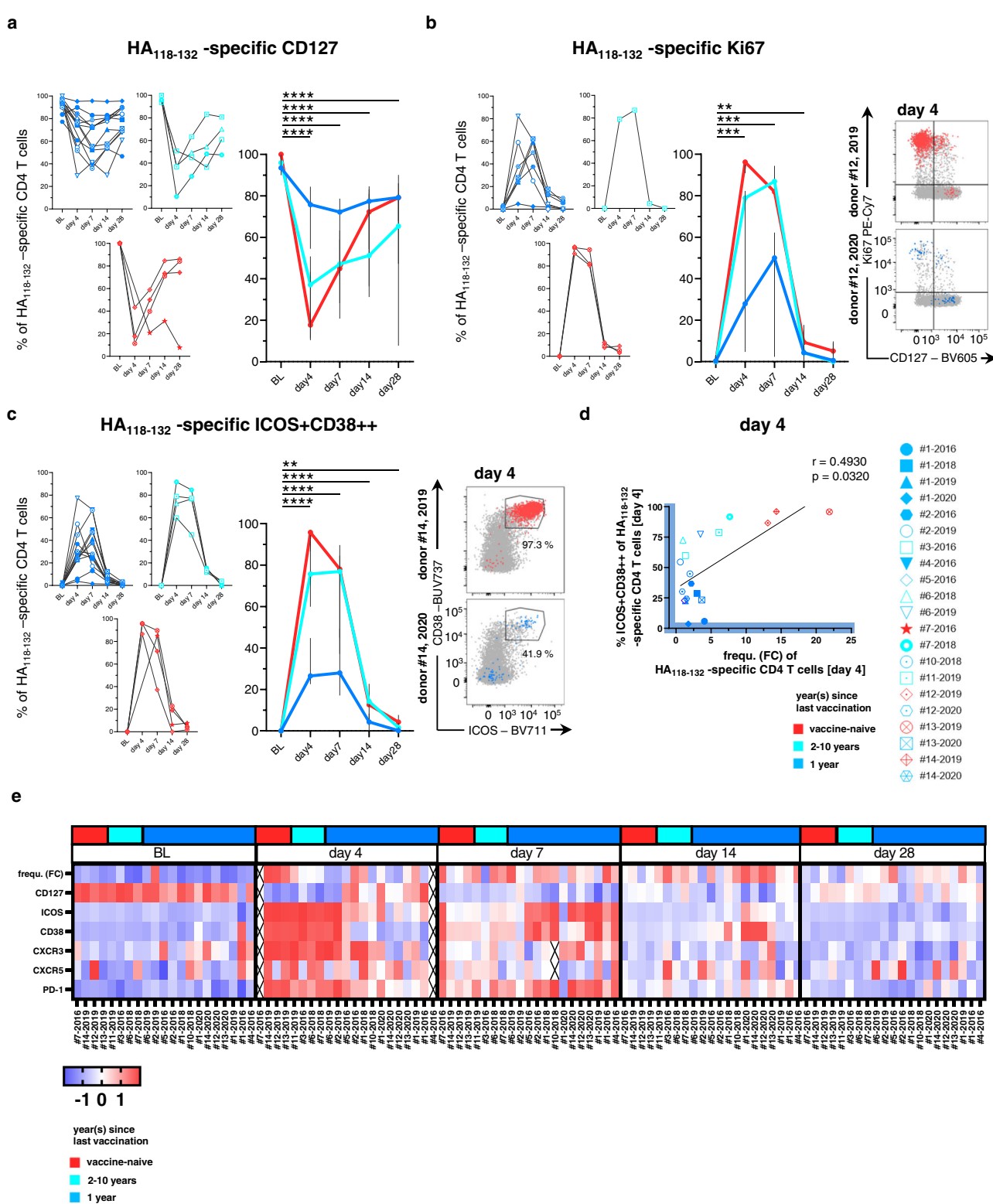

**Fig. 2 HA$_{118-132}$-specific CD4 T cells are highly activated and proliferate upon vaccination and the individual dynamics are associated with vaccination history. a–c** Expression of CD127, Ki67, and ICOS+CD38++ on HA$_{118-132}$-specific CD4 T cells at the indicated time points. Responses are subgrouped by vaccination history. The summary plot shows the median with 95% confidence interval within the three subgroups (for CD127 and ICOS+CD38++: BL, day 7–day 28 $n = 21$, day 4 $n = 19$; Ki67 BL—day 28 $n = 11$). One symbol represents response at the indicated time point. Representative dot plots in **b**, **c** show an example for a vaccine-naive donor (red) and a successive vaccinated donor (blue), gray points show the bulk population. **d** Spearman correlation of ICOS, CD38 co-expressing HA$_{118-132}$-specific CD4 T cells at day 4 with frequency (FC) of HA$_{118-132}$-specific CD4 T cells at day 4. **e** Expression dynamics of markers during HA$_{118-132}$-specific CD4 T cell responses ($n = 21$). Heatmap coloring represents the z score of every marker (MFI) calculated among the time points of an individual donor response. Responses are aligned according to vaccination history. **a–c** Two-tailed Wilcoxon matched-pairs signed rank test with Bonferroni correction was used for multiple between-group comparisons. All time points were compared to BL. A p value of 0.0125 was considered statistically significant. **$p < 0.01$, ***$p < 0.001$, ****$p < 0.0001$. **d** Two-tailed Spearman correlation test with a confidence interval of 95%. *$p < 0.05$.

on day 4 and a group with activation predominantly on day 7. In both groups, changes largely returned to baseline levels between day 14 and day 28 post vaccination (Fig. 2e).

**HA$_{118-132}$-specific CD4 T cells display two distinct activation signatures.** Based on the phenotype of HA$_{118-132}$-specific CD4 T cells 4 days after vaccination, we performed hierarchical clustering of the individual responses in order to determine whether the vaccination history was indeed the strongest separator between individual responses. Using an unbiased approach, however, the vaccine cohort separated into two groups (Fig. 3a). The group including individuals displaying a strong HA$_{118-132}$-specific CD4 T cell activation at day 4 after vaccination was termed "early" (green), while the group including those patients with delayed or absent HA$_{118-132}$-specific CD4 T cell activation upon vaccination was termed "late" (black) (Fig. 3a). Importantly, vaccination history was closely but not exclusively associated with "early" and "late" HA$_{118-132}$-specific CD4 T cell activation. Indeed, all vaccine responses classified as "late" were observed in individuals who had received the seasonal vaccine in the previous year, whereas most "early" responses were observed in individuals who had received their first influenza vaccination or had not received a vaccination in the previous year (Fig. 3a). This observation is exemplified by individuals #12, #13, and #14 who received their first vaccination in 2019 and displayed an "early" activation pattern on HA$_{118-132}$-specific CD4 T cells and a "late" activation pattern following their second vaccination in 2020 (Supplementary Fig. 4a). Among all the markers that were analyzed, CD38 and ICOS expression 4 days after vaccination most strongly discriminated between the "early" and the "late" groups (Fig. 3b and Supplementary Fig 4b). In addition, definite classification of vaccine responses into an "early" and a "late" group was even independently possible by analyses of expression levels of either marker (Fig. 3b). Next, we visualized the HA$_{118-132}$-specific CD4 T cells at baseline and on days 4 and 7 after vaccination using dimension reduction with t-distributed stochastic neighbor embedding (tSNE) analyses. These analyses revealed the presence of four distinct T cell clusters within the HA$_{118-132}$-specific CD4 T cell population (Fig. 3c). While cluster 1 represents cells with a resting memory phenotype (CD127+, TCF-1+), cluster 2 represents strongly activated, proliferating Tfh-like cells (CD38+, ICOS+, Ki67+, CXCR5+, PD-1+), cluster 3 represents strongly activated and proliferating cells with less Tfh cell characteristics (CD38+, ICOS+, Ki67+), and cluster 4 includes cells with a moderately activated, effector phenotype (CD127low, CD38low, T-bet high, see also corresponding heatmap). At baseline, HA$_{118-132}$-specific CD4 T cells independent of the vaccination history are located in the memory cluster 1. On day 4, clusters 2 and 3 were populated with cells that predominantly were derived from individuals who had not received a vaccination in the previous year (red and turquoise) or who displayed an "early" activation phenotype (green), depending on the group stratification. In contrast, cells from recently vaccinated

individuals (blue) and those that displayed a "late" activation phenotype (black) maintained their cluster 1 localization. On day 7, the effector cluster 4 was the most populated cluster, including cells from all groups. Thus, tSNE analyses reveal the presence of two distinct but transient clusters of highly activated and proliferative HA$_{118-132}$-specific CD4 T cells (clusters 2 and 3) that are largely specific and unique to cells derived from individuals with an "early" activation response. In addition, cells from individuals with a "late" response maintain their memory characteristics (cluster 1) or convert directly to the moderately activated, effector phenotype cluster 4 on day 7 without populating clusters 2 and 3. Collectively, these data demonstrate that vaccine-associated HA$_{118-132}$-specific CD4 T cell responses can exhibit two distinct activation patterns.

**Emergence of activated HA$_{118-132}$-specific cTfh cells.** As early activated effector cells are known to increase their glycolytic activity to balance the high energetic demand[36–39], we analyzed the uptake of 2-NBDG, a fluorescent tracer for glucose uptake. This was particularly relevant as it has previously been suggested that secondary expansion of memory T cells in response to vaccination occurs independently of glycolysis[40]. Within all HA$_{118-132}$-specific CD4 T cells, we observed the tendency of elevated 2-NBDG uptake after vaccination (Fig. 4a). More detailed analysis at the level of the differentially activated subpopulations of HA$_{118-132}$-specific CD4 T cells (Fig. 4b) revealed significantly elevated 2-NBDG uptake within ICOS+CD38++ cells compared to ICOS−CD38+ and double-negative HA$_{118-132}$-specific CD4 T cells (Fig. 4c and Supplementary Fig. 5c) indicating a shift of their metabolic demands as activated, proliferating cells. In previous studies, ICOS and CD38 upregulation was associated with the presence of a cTfh cell phenotype on bulk CD4 T cells after influenza vaccination[23,27] (Supplementary Fig. 5a). This observation could be confirmed on HA$_{118-132}$-specific CD4 T cells (Fig. 2c and Supplementary Fig. 5b). Indeed, cells with a CXCR5+PD-1+ (cTfh) phenotype were most frequently detected within the ICOS+CD38++ subset (Fig. 4d). As expected, based on the cluster analysis in Fig. 3c, individuals who have not been vaccinated in the previous year displayed a significantly stronger emergence of HA$_{118-132}$-specific cTfh cells compared to those who had recently been vaccinated (Fig. 4e). Next, we analyzed the expression of the transcription factor TOX. Members of the TOX family (especially TOX and TOX2) were recently identified as critical regulators of Tfh development in mice upstream of Bcl6[25]. TOX expression was significantly higher in cTfh vs. non-cTfh HA$_{118-132}$-specific CD4 T cells early after vaccination (Fig. 4f and Supplementary Fig. 4c), suggesting that TOX might be involved in the emergence of cTfh cell responses within HA$_{118-132}$-specific CD4 T cells upon vaccination. Taken together, the ICOS+CD38++ influenza-specific CD4 T cell population that shapes the "early" activation response after seasonal influenza vaccination is characterized by high metabolic

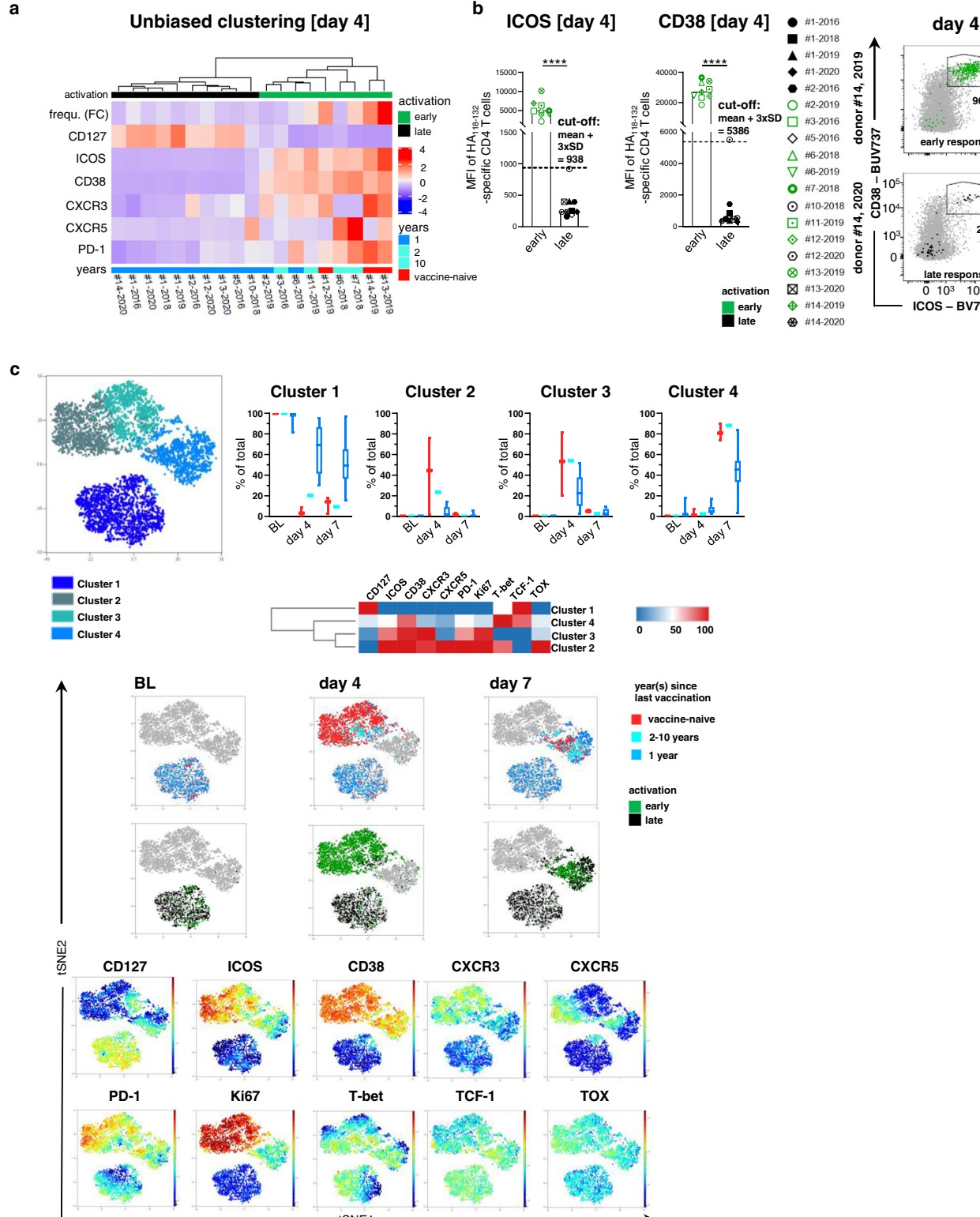

demands and phenotypic and transcriptional markers of Tfh differentiation (Supplementary Fig. 5d).

**Early CD4 T cell activation precedes HA-specific immunoglobulin G (IgG) induction.** The strong interconnection between early influenza-specific CD4 T cell activation and the emergence of a Tfh signature on day 4 raises the question whether these

signatures are associated with vaccine-elicited antibody titers as has recently been shown on the bulk CD4 level[10,11,30]. In order to confirm these results on the protein-specific level, we assessed antibody titers against the respective seasonal HA recombinant protein and the entire vaccine by enzyme-linked immunosorbent assay (ELISA) at baseline and 28 days after vaccination. Since the HA protein from A/Singapore/INFIMH-16-0019/2016 was not available, the HA protein from A/Hong Kong/4801/2014 with a

**Fig. 3 Longitudinal analysis of HA$_{118-132}$-specific CD4 T cell responses shows two distinct activation signatures after seasonal influenza vaccination. a** Heatmap of z scores calculated per marker based on MFI of marker expression levels on HA$_{118-132}$-specific CD4 T cells and frequency (FC) of HA$_{118-132}$-specific CD4 T cells from all analyzed responses on day 4 (n = 19) with unbiased clustering of the responses. **b** Cut-off value of CD38 and ICOS expression on HA$_{118-132}$-specific CD4 T cells day 4 (mean MFI + 3× standard deviation (SD) of the late responder group) for discrimination of early (green n = 9) and late responses (black n = 10). Representative dot plots show co-expression of ICOS+CD38++ on HA$_{118-132}$-specific CD4 T cells for an early and a late response. **c** t-Distributed Stochastic Neighbor Embedding (tSNE) analysis with FlowSOM clustering was performed of HA$_{118-132}$-specific CD4 T cells based on polychromatic flow cytometric data (panel 2) including all responses (vaccination history: red n = 3, turquoise n = 1, blue n = 7, activation: green n = 6, black n = 5) at the indicated time points. The frequency of HA$_{118-132}$-specific CD4 T cells was analyzed within the clusters at the indicated time points (box plots). The heatmap summarizes the phenotypical characteristics of clusters 1–4. The color scaling of the tSNE plots below indicates the expression levels of the indicated makers. Within tSNE dot plots, the expression levels of individual surface marker expression is colored according to channel fluorescence intensity. Color scales are denoted adjacent to tSNE dot plots. Bars represent the median. One symbol represents one response at the indicated time point. **b** Two-tailed Mann–Whitney U-test was used for comparisons between the two groups. ****p < 0.0001.

99% sequence homology was used to analyze samples from the 2018/2019 vaccine season. However, testing all samples against all three HA proteins revealed almost identical results with the exception of the 2019 samples that showed lower FC when tested against the HA protein from A/Hong Kong/4801/2014 compared to the HA protein from A/Kansas/14/2017 (which corresponds to the 2019 vaccine strain and has a 98% homology to the HongKong strain) (Supplementary Fig. 6e) Most individuals displayed an increase in HA-specific antibodies from baseline to day 28 with few individuals from the recently vaccinated group showing stable or even declining titers (Fig. 5a). In contrast, IgG levels against the entire vaccine increased in all vaccinees (Supplementary Fig. 6a, b). The FC of HA-specific antibodies from baseline to day 28 was significantly higher in individuals who had not been vaccinated in the previous year (Fig. 5b and Supplementary Fig. 6c). However, the presence of an "early" activation phenotype on HA$_{118-132}$-specific CD4 T cells was an even stronger predictor of an increase in HA-specific antibodies (Fig. 5b). HA-specific IgG levels on day 28 were similar in both groups (Fig. 5c). In line with the association between an "early" activation phenotype and strong IgG induction, the percentage of HA$_{118-132}$-specific ICOS+CD38++ CD4 T cells at day 4 as well as the percentage of HA$_{118-132}$-specific cTfh cells at day 4 correlated with the FC of HA-specific antibodies (Fig. 5d and Supplementary Fig. 6d), suggesting a direct link between early influenza-specific CD4 T cell activation and antibody induction. In contrast, the FC of the HA$_{118-132}$-specific CD4 T cell frequency at day 4 showed no correlation with the FC of HA- or vaccine-specific antibodies (Fig. 5d and Supplementary Fig. 6d), indicating that phenotypic changes more specifically associate with antibody induction than the FC of the HA$_{118-132}$-specific CD4 T cell frequency. These data implicate the induction of antibody responses to the seasonal influenza vaccination with the "early" virus-specific CD4 T cell activation dynamics.

**Baseline parameters associated with HA-specific IgG induction.** In order to decipher which immunological parameters determine the activation and expansion of the HA$_{118-132}$-specific CD4 T cells, we first analyzed which baseline parameters are associated with an "early" activation phenotype. For this analysis, we included the influenza-specific CD4 T cell and the humoral compartment since both, T cell intrinsic mechanisms and binding of the vaccine antigens by pre-existing antibodies may influence the virus-specific CD4 T cell response. Among the different baseline parameters shown in Fig. 6a (BL parameters stratified for vaccine history are shown in Supplementary Fig. 8), pre-existing HA-specific IgG levels and the expression levels of CD127 on HA$_{118-132}$-specific CD4 T cells showed a significant difference between the groups. Next, we aimed to analyze how the central parameters at baseline, day 4, and day 28 are interconnected and performed unbiased clustering, which showed a separation by

vaccination history as well as by early and late responders (Fig. 6b). These parameters were subsequently included in a correlation matrix in order to analyze whether the relevant baseline parameters correlate among themselves (framed in purple), with HA$_{118-132}$-specific CD4 T cell activation and cTfh differentiation at day 4 (green) and HA-specific IgG induction at day 28 (yellow). Among baseline parameters, pre-existing HA-specific IgG levels showed a reverse correlation with the percentage of CD127 expressing HA$_{118-132}$-specific CD4 T cells and a positive correlation with the percentage of HA$_{118-132}$-specific CD4 T cells (Fig. 6c and Supplementary Fig. 7a). All three baseline parameters correlated significantly with the percentage of HA$_{118-132}$-specific ICOS+CD38++ CD4 T cells and—with the exception of baseline CD127 expression—with a cTfh phenotype on HA$_{118-132}$-specific CD4 T cells on day 4 (Supplementary Fig. 7b, c). None of the baseline parameters correlated with the FC in HA$_{118-132}$-specific CD4 T cell frequency from baseline to day 4 (Fig. 6c). HA-specific IgG levels and the percentage of CD127 expression on HA$_{118-132}$-specific CD4 T cells at baseline significantly correlated with the vaccine-elicited HA-specific IgGs (displayed as FC from baseline to day 28) (Supplementary Fig. 7d). In contrast, the baseline frequency of HA$_{118-132}$-specific CD4 T cells did not correlate with the IgG FC (Fig. 6c). When analyzing IgG responses against the entire vaccine (not only HA proteins), the correlation matrix largely confirmed the above-described correlations (Supplementary Fig. 9). Corresponding receiver operating characteristic (ROC) analyses revealed a predictive potential of baseline parameters not only for the HA$_{118-132}$-specific CD4 T cell response at day 4 (early vs. late) but also for high vs. low antibody induction (Supplementary Fig. 10). Collectively, these results suggest that pre-existing antibodies and the frequency of CD127 expression on T cells both targeting the same influenza protein are central determinants for the course of the response against successive seasonal influenza vaccination that is characterized by strong CD4 T cell activation and followed by an increase in antibody titers.

## Discussion
Protection from disease after influenza vaccination is predominantly mediated by neutralizing antibodies that are generated in a T cell-dependent maturation process. Since Tfh cells have been established as a separate CD4 T cell lineage that is specialized in providing help to B cells in the context of affinity maturation, several studies have investigated CD4 T cell responses following seasonal influenza vaccination with a special focus on cells with a Tfh phenotype (i.e. co-expression of PD-1 and CXCR5). Importantly, several studies have shown a direct relationship between the frequency of activated cTfh cells following vaccination and influenza vaccine-induced antibody responses[10,11,30,41]. However, while antibody responses are typically analyzed in an antigen-specific fashion, activation or

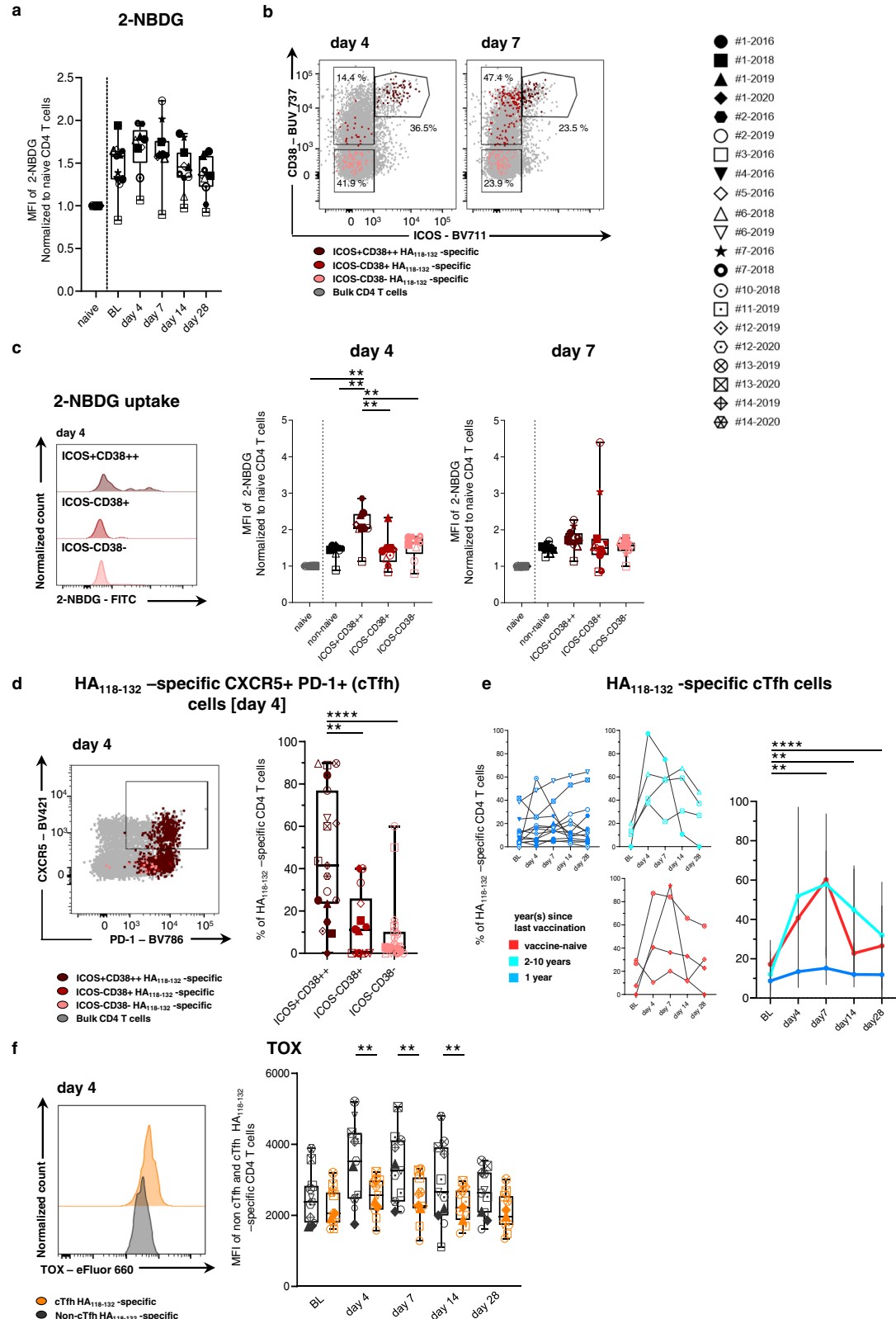

expansion patterns of their T cell counterparts have mostly been measured on bulk populations[10,11,27–30,41], hindering conclusions on protein-specific T cell–B cell interactions. Indeed, analyses of influenza-specific CD4 T cells after vaccination have been hampered by three aspects: (1) The rapid development of viral variants due to antigenic drift and antigenic shift, (2) HLA diversity in humans, and (3) the small frequencies of antigen-specific CD4

T cells. In order to overcome these hurdles, we (1) identified an HLA-DRB1*01:01–MHC class II–restricted HA-derived epitope ($HA_{118-132}$) from an H3N2 strain that was conserved throughout the 2016/2017, 2018/2019, 2019/2020, and/ 2020/2021 seasons, (2) performed HLA typing in all participating individuals, and (3) used bead-based tetramer enrichment techniques to recover sufficient numbers of $HA_{118-132}$-specific CD4 T cells for direct

**Fig. 4 Highly activated ICOS and CD38 co-expressing HA$_{118-132}$-specific CD4 T cells with altered metabolic demands show a Tfh phenotype early after vaccination. a** Glucose uptake of HA$_{118-132}$-specific CD4 T cells was determined using the fluorescent tracer 2-NBDG. Longitudinal analysis of 2NBDG (nMFI) at the indicated time points ($n = 11$). **b** Representative dot plots of CD38 and ICOS expression on HA$_{118-132}$-specific CD4 T cells showing different activated populations upon vaccination (ICOS+CD38++ = dark red, ICOS−CD38+ = red, ICOS−CD38− = pink). **c** nMFI of 2-NBGD is shown within the different activated HA$_{118-132}$-specific populations with representative histograms (day 4 $n = 9$; day 7 $n = 11$). **d** Analysis of CXCR5+PD-1+ (cTfh) phenotype within the indicated HA$_{118-132}$-specific populations (defined in **b**) on day 4 (ICOS+CD38++ $n = 19$, ICOS−CD38+ $n = 14$, ICOS−CD38− $n = 19$). **e** Frequency of cTfh cells within HA$_{118-132}$-specific CD4 T cells at the indicated time points. Responses are subgrouped by vaccination history (red $n = 4$, turquoise $n = 4$, blue $n = 13$). The summary plot (right) represents the median of the subgroups. Error bars show the 95% confidence interval. **f** TOX expression (MFI) within cTfh (marked in orange) and non-cTfh (marked in dark gray) HA$_{118-132}$-specific CD4 T cells at the indicated time points ($n = 11$). One symbol represents one response at the indicated time point. **a–e** Two-tailed Wilcoxon matched-pairs signed rank test with Bonferroni correction for multiple comparisons was used for between-group comparisons. **f** Two-tailed Mann–Whitney $U$-test was used for comparisons between two groups. In **c**, **d**, all populations were compared to the ICOS+CD38++ population. $p$ values of 0.01 (**) (**a**, **e**), 0.017 (*) (**c**), and 0.025 (*) (**d**) were considered as statistically significant. **$p < 0.01$, ****$p < 0.0001$. nMFI normalized mean fluorescence intensity.

ex vivo analyses. Importantly, and in contrast to studies analyzing bulk CD4 T cells or using activation-induced markers, this approach offers the possibility to longitudinally track HA$_{118-132}$-specific CD4 T cells from the resting state at baseline throughout the vaccine-induced activation. In contrast to CD4 T cells targeting a mutated epitope that has been used in previous studies of vaccine-induced CD4 T cell activation (HA$_{306-318}$), HA$_{118-132}$-specific CD4 T cells showed strong signs of activation and expansion after vaccination, clearly demonstrating antigen availability as a prerequisite for vaccine-induced CD4 T cell activation (Fig. 2 and Supplementary Fig. 2). However, the observation of two distinct response patterns, an "early" activation and a "late/absent" activation, clearly demonstrates that antigen recognition alone does not determine the vaccine response. Indeed, it has previously been shown that repeat influenza vaccination reduces the antibody induction and impairs the affinity maturation of influenza-specific antibodies[4–9]. This observation has also been linked to the activation of bulk cTfh cells 7 days after vaccination that has been shown to be lower in those individuals who had received the vaccine in the previous years compared to those who did not[10,11,27–30,41]. However, it remained unclear whether this association can be recapitulated on an antigen-specific level and whether the analysis of influenza-specific T cells might provide additional insights regarding the dynamics of these interactions. One novel insight was the identification of two highly activated T cell clusters that were evident in a subset of vaccines on day 4 after vaccination and were characterized by strong ICOS and CD38 expression, with (cluster 2) or without (cluster 3) a cTfh signature (Fig. 3c). These clusters were transient and not present on day 7, a time point at which many studies have analyzed vaccine-elicited T cell responses. The presence of these clusters was largely restricted to patients with an "early" response, which was in turn associated with a strong induction of antibodies targeting the same protein. Similar to previous observations on the bulk level[10,11], the individual vaccine history was closely associated with the presence of the "early" activation pattern and most vaccinees displaying strong CD4 T cell activation had not received a vaccine in the previous year (Fig. 3a). However, some individuals with repeat vaccination who displayed an "early" activation subsequently showed a strong increase in antibody responses, making the presence of an "early" activation signature on HA$_{118-132}$-specific CD4 T cells a better predictor of an increase of HA-specific antibodies than the vaccine history (Fig. 5b).

The antigen-specific approach also allowed us to analyze the composition of the HA$_{118-132}$-specific CD4 T cell pool regarding follicular T helper lineage commitment and the defining transcription factors. Indeed, several transcription factors such as Bcl6 and TCF-1 are involved in the Tfh program. However, Bcl6 is not detectable in cTfh cells[18,19,23] and TCF-1 expression is also involved in memory formation[26] and heavily downregulated after

activation and is therefore not suited to confirm Tfh lineage commitment in the immediate post-vaccination scenario. More recently, TOX and TOX2 have been shown to support the Tfh program upstream of Bcl6 in mice[25]. Whether TOX is also expressed in antigen-specific Tfh cells in humans has not been analyzed. The observation that TOX expression is induced after vaccination most strongly on cells with a cTfh phenotype suggests that TOX might be useful as a transcriptional surrogate of Tfh lineage commitment in cTfh cells (Fig. 4f).

Collectively, our data clearly suggest that activation of vaccine-specific CD4 T cells and acquisition of Tfh characteristics are important prerequisites for the emergence of vaccine-elicited antibodies and that this is also connected to an individual's vaccine history. However, it remains to be demonstrated which mechanisms prevent CD4 T cell activation in those who receive repeat vaccinations. One possible explanation could be a refractory state of T cells after repeat circles of re-activation, which would be supported by the expression patterns of CD127, the IL-7-receptor alpha-chain. CD127 expression identifies T cells with memory characteristics[35,42] as signaling via CD127 promotes survival and maintenance of long-lived memory T cells[43]. In chronic hepatitis C virus (HCV) infection for example, the ability of HCV-specific CD8 T cells to proliferate is closely associated with the expression of CD127. Indeed, HCV-specific CD8 T cells can be separated into memory-like cells with retained functional capacities that express high levels of CD127 expression and terminally exhausted CD8 T cells with lower CD127 expression[44,45]. Similarly, in our cohort, high CD127 expression at baseline was associated with strong activation 4 days after vaccination (Fig. 6a). Subsequently, CD127 was downregulated on proliferating cells[46] and CD127-expressing cells again dominated the specific CD4 T cell population 28 days after vaccination when the antigen was no longer available[47] (Fig. 2). Thus, low CD127 expression because of repeat antigenic stimulation might be one factor responsible for the different activation kinetics of HA$_{118-132}$-specific CD4 T cells. It is worth noting in this context that HA$_{118-132}$-specific CD4 T cells displayed an antigen-experienced memory phenotype in all individuals, regardless of the vaccination history, suggesting that all individuals had been exposed to an influenza strain carrying this HA peptide (Fig. 2 and Supplementary Fig. 3). Thus, the differences between early and late responses are not merely due to initial priming vs. secondary challenge. Another potential explanation for the different activation signatures could be the pre-existing antibody levels. Indeed, compared to individuals with a vaccine-naive history, those who had recently been vaccinated had significantly higher titers of pre-existing vaccine-specific IgGs at baseline (Fig. 6 and Supplementary Fig. 8). One possible explanation that has been brought forth to explain ineffective seasonal vaccination is the concept of the original antigenic sin, which describes the phenomenon that antibodies targeting epitopes of the first strain an individual has been exposed to are predominantly boosted after vaccination at the expenses of antibodies targeting novel epitopes[48,49]. However, this concept may

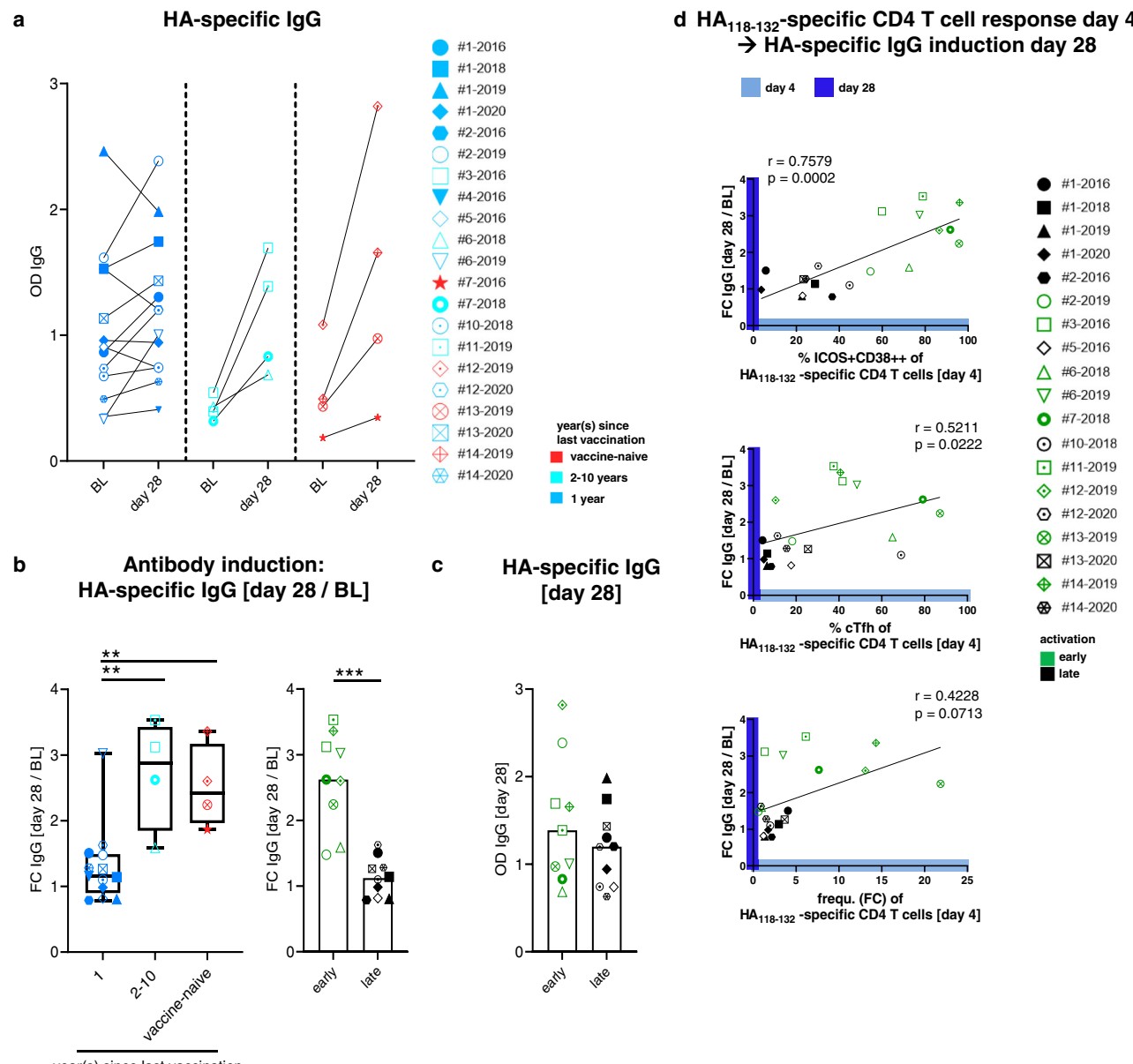

**Fig. 5 HA$_{118-132}$-specific CD4 T cell activation and Tfh phenotype on day 4 correlate with the increase of HA-specific IgG after vaccination. a** HA-specific IgG plasma levels, quantified by ELISA, are shown at the indicated time points. Responses are subgrouped by vaccination history. Two-tailed Wilcoxon matched-pairs signed rank test was used for between-group comparisons. **b** HA-specific IgG induction as fold change (FC: day 28/BL) grouped by the vaccination history (red $n = 4$, turquoise $n = 4$, blue $n = 13$) and by early and late responses (green $n = 9$, black: $n = 10$). **c** HA-specific IgG levels [day 28] grouped by early and late responses (green $n = 9$, black: $n = 10$). Two-tailed Mann–Whitney $U$-test was used for comparisons between two groups. ***$p < 0.001$. **b** Two-tailed Mann–Whitney $U$-test was used with Bonferroni correction for comparisons between two groups. $p$ values of 0.025 was considered as statistically significant: **$p < 0.01$. **d** Spearman correlations of ICOS, CD38 co-expressing HA$_{118-132}$-specific CD4 T cells at day 4, CXCR5, and PD-1 co-expressing HA$_{118-132}$-specific CD4 T cells at day 4 and frequency (FC) of HA$_{118-132}$-specific CD4 T cells (day 4/BL) with FC of HA-specific IgG [day 28/BL] ($n = 19$). Responses are subgrouped by early and late responders (green $n = 9$, black $n = 10$). Two-tailed Spearman correlation test with a confidence interval of 95%. *$p < 0.05$, **$p < 0.01$, ***$p < 0.001$. One symbol represents one response at the indicated time point.

not apply in our study focusing on T cells targeting a conserved epitope that are differentially activated in repeated vaccinations even within the same individuals (Supplementary Fig. 4). Alternatively, pre-existing antibodies have been suggested to mask relevant epitopes that could subsequently prevent the stimulation of specific B cells[50,51]. Thus, it is equally conceivable that such a mechanism might result in insufficient uptake of antigens by antigen-presenting cells (APCs) and subsequently reduced presentation via MHC class II to influenza-specific CD4 T cells. Furthermore, IgG-opsonized antigens (immune complexes) influence antigen availability and processing by

binding to Fcγ receptors on APCs and B cells. Thereby, the different route of antigen uptake is a key determinant for further antigen processing and presentation[52–54]. "Negative interference" of pre-existing antibodies with antigen availability uptake and presentation might also be involved in the context of the "antigenic distance hypothesis"[8,55]. Indeed, this hypothesis was developed based on the observation that, in certain contexts, repeat vaccinations suffer from limited efficacy and postulates that a small antigenic distance between two vaccine strains results in limited vaccine efficacy, especially when the season's epidemic strain is antigenically more distant[56]. While the

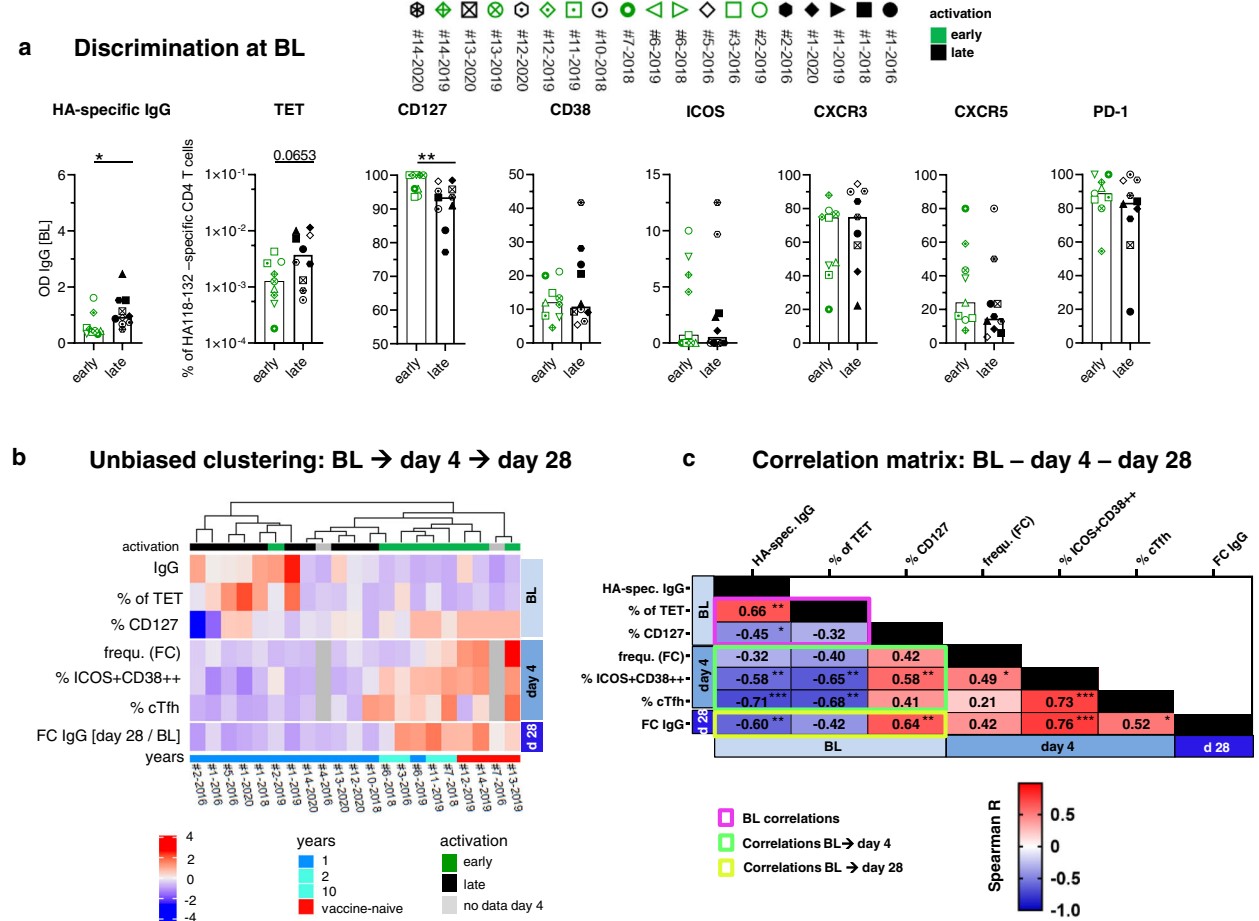

**Fig. 6 BL parameters that influence the dynamics of HA$_{118\text{-}132}$-specific CD4 T cell activation and HA-specific IgG induction after seasonal vaccination.**
**a** Analysis of BL parameters to discriminate early and late responses. Bars represent the median. One symbol represents one response at the indicated time point. Two-tailed Mann–Whitney U-test was used for comparisons between two groups. ns > 0.05, *p < 0.05, **p < 0.01 (green n = 9, black n = 10). **b** Heatmap of z scores calculated per parameter with unbiased clustering to visualize correlations of parameters from BL to day 4 and day 28 (n = 21 responses). **c** Correlation matrix with Spearman r value from BL to day 4 and day 28 (n = 19 responses). **b**, **c** Two-tailed Spearman correlation test with a confidence interval of 95%. *p < 0.05, **p < 0.01, ***p < 0.001.

clinical relevance of this theory with respect to vaccine efficacy and translation into protective efficacy remains to be proven, the concept of "negative interference" could nevertheless explain the weak HA$_{118\text{-}132}$-specific CD4 T cell activation and subsequent impaired antibody induction after repeat vaccination. In addition, the inverse correlation between pre-existing antibodies and the antibody induction may suggest that a certain target level of antibodies specific for a given antigen is centrally regulated[11,30,57]. This assumption would be supported by our observation that antibody levels were similar between groups at day 28 (Fig. 5c), suggesting that an immune-rheostat strives for a certain target level and prevents immune activation in response to the vaccine if the target level is already reached. In contrast to this hypothesis, it has recently been shown that individuals receiving severe acute respiratory syndrome coronavirus 2 vaccination after natural infection show stronger antibody responses than those with either natural infection or vaccination alone[58–61]. The sequence of natural infection followed by vaccination results in a state termed "hybrid immunity" and the higher antibody titers observed in these individuals has been attributed to the idea that the host perceives repeat antigenic exposure as an increased threat and subsequently strives for higher antibody titers[62].

Collectively, our data indicate that pre-existing vaccine-specific immunity and to a lesser degree an individual's vaccination history determine the responsiveness to seasonal influenza vaccination on the levels of vaccine-specific CD4 T cells and that CD4 T cell activation is required for the emergence of corresponding antibodies. The failure of repeat vaccinations to induce CD4 T cell activation, despite the presence of their cognate antigen, should be considered in the design of novel vaccination strategies against seasonal viruses and requires additional research to identify the underlying mechanisms.

## Methods

**Design**. A total of 22 healthy donors, who were vaccinated with seasonal influenza vaccines in the time periods from 2016 until 2020, were enrolled in the study (Table 2 and Fig. 1b). Samples were obtained at BL (before vaccination), day 4, day 7, day 14, and day 28 after vaccination. In selected donors, additional samples were taken at day 2, day 10, and 28 days after the last sample on day 28 (follow-up 28) (Fig. 1a). Donor characteristics are included in the Table 1. The individual vaccination responses are grouped by the donors' self-reported vaccination history. Antigen-specific CD4 T cell analysis was performed with samples from 12 DRB1*01:01-positive vaccinated individuals. HLA-typing was performed by next-generation sequencing using commercially available primers (GenDx, Utrecht, The Netherlands). Samples were run on a MiSeq system. NGSengine® Software (GenDx) was used for data analysis.

**Ethics**. This study was approved by the Ethics Committee of the Albert-Ludwigs-University Freiburg (474/14, 201/17) and written informed consent was obtained from all blood donors before enrollment in the study.

**Preparation of peripheral blood mononuclear cells (PBMCs) for staining**. ETDA-anticoagulated venous blood samples were collected in Monovette tubes

(Sarstedt, Nuembrecht, Germany). PBMCs were isolated by density gradient centrifugation using Pancoll (PAN-Biotech, Aidenbach, Germany) and washed with phosphate-buffered saline (PBS, PAN-Biotech). The cells were cryopreserved at −80 °C. Thawing of PBMCs was performed in complete medium (RPMI1640 supplemented with 10% fetal calf serum, 100 U/ml penicillin, 100 µg/ml streptomycin, 10 mM HEPES; Fisher Scientific GmbH, Schwerte, Germany).

**Seasonal influenza vaccination and tetramer for T cell analysis.** Table 2 lists the seasonal vaccines in 2016–2020 with the associated influenza A strains of H3N2. Phycoerythrin (PE)-labeled MHC class II tetramers of HLA-DRB1* 01:01 Influenza (Flu)-derived epitope HA$_{118-132}$ (aa-sequence: VPDYASLRSLVASSG) and epitope HA$_{306-318}$ (aa-sequence: PKYVKQNTLKLAT) were obtained from MBL, Woburn, United States (Supplementary Table 1).

**Flow cytometry.** PBMCs were surface stained or fixed and permeabilized to stain intranuclear targets with specific antibodies (Supplementary Table 2 and see "Multiparametric flow cytometry for T cell analysis") Cells were plated in 96-well-V-bottom plates and washed two times and resuspended in staining buffer (PBS + 1% fetal calf serum). Surface antibody mixtures were added followed by 40 min incubation at room temperature (RT). Fixable Viability dye$^{eFluor780}$ (eBioscience, Germany) was used for dead cell exclusion. Cells were then washed again and fixed in fixation buffer (PBS + 2% (w/v) paraformaldehyde). For intranuclear staining, the FoxP3/Transcription Factor Staining Buffer Set (eBioscience, Germany) was used after surface staining according to the manufacturer's instructions. Cells were resuspended in fixation buffers.

Stained cells were analyzed with an LSR Fortessa (BD Biosciences) and multiparametric flow cytometric data were collected by the FACSDiva Software (Becton Dickinson). FlowJo 10.0.7 (LLC, BD Life Sciences, Ashland, OR, USA) was applied for the analysis of flow cytometric data. Gating strategies are shown in Supplementary Fig. 11.

**Magnetic bead-based enrichment of antigen-specific CD4 T cells.** Enrichment of virus-specific CD4 T cells was adapted from the method as described previously[63]. In brief, 1–2 × 10$^7$ PBMCs of DRB1*01:01-positive donors were labeled with PE-coupled peptide-loaded HLA class II tetramers for 30 min. Tetramer specifications are listed in Supplementary Table 1. Subsequent enrichment was performed with anti-PE beads using MACS technology (Miltenyi Biotec, Germany) according to the manufacturer's protocol. Enriched influenza-specific CD4 T cells were used for multiparametric flow cytometric analysis. Samples with less than five antigen-specific CD4 T cells (considered below limit of detection) were excluded from the final analysis. The frequency of influenza-specific CD4 T cells was calculated as follows: absolute number of influenza-specific CD4 T cells (enriched sample) divided by the absolute number of CD4 T cells (pre-enriched sample) × 200.

**Analysis of glucose uptake.** For monitoring glucose uptake, enriched influenza-specific CD4 T cells were incubated with 2-NBDG (100 µM) for 20 min at 37 °C in complete medium. After washing in complete medium, cells were surface stained as explained above.

**Multiparametric flow cytometry for T cell analysis.** The following antibodies were used for multiparametric flow cytometry: anti-CD4-FITC (RPA-T4, 8037703, 1:50, Cat#555346), anti-CCR7-BUV395 (3D12, 43026, 4:100, Cat#563977), anti-CXCR5-APC (RF8B2, 558113, 1:50, Cat#356908), anti-PD-1-BV786 (EH12.1, 1013122, 3:100, Cat#563789), anti-CD134(OX40)-PE-Cy7 (ACT-35, 65008, 3:100, Cat#563663), anti-CD38-BUV737 (HB7, 8339566, 1:200, Cat#564686), anti-ICOS-BV711 (DX29, 7164540, 1:100, Cat#563833), anti-CD127-BV421 (HIL-7R-M21, 9346341, 3:100, Cat#562436), anti-CCR6-BV605 (11A9, 9065844, 3:100, Cat#562724), anti-T-bet-PE-CF594 (O4-46,93533305, 3:100, Cat#562467) (BD Biosciences), anti-CD4-AlexaFluor700 (RPA-T4, 300526, 1:200, Cat#300526), anti-CXCR5-BV421 (J252D4, B252332, 1:100, Cat#562747), anti-CXCR3-BV510 (G025H7, B293007, 3:100, Cat#353726), anti-CD127-BV605 (A019D5, B320426, 3:100, Cat#351334), anti-CD27-PE-Dazzle594 (M-T271, B291287, 1:400, Cat#356422), anti-Ki67-PE-Cy7 (Ki67, B269861, 1:200, Cat#350525) (BioLegend), anti-TCF-1-AlexaFluor488 (C63D9, 1:100, Cat#6444) (Cell Signaling), anti-CD14-APC-eFluor780 (61D3, 1:100, Cat#47-0149-42), anti-CD19-APC-eFluor780 (HIB19, 1:100, Cat#47-0199), anti-TOX-eFluor660 (TRX10, 1:100, Cat#50-6502) (eBioscience), anti-CD45RA-PerCP-Cy5.5 (HI100, 3:100, Cat#45-0458-42), 2-NBDG (20 mM) (1:200, Cat#N13195) (Invitrogen). For live/dead discrimination, a fixable Viability Dye (APC-eFluor780 1:200, Cat#65-0865) (eBioscience) was used.

**Enzyme-linked immunosorbent assay.** For the quantification of vaccine-specific IgG levels and HA recombinant IgG levels, aliquots of plasma samples were thawed prior to ELISA procedure. To determine influenza-specific IgG levels, Nunc MaxiSorp Plates (VWR, Bruchsal, Germany) were coated with 50 µL/well of seasonal influenza vaccines (diluted 1:100) and incubated at 4 °C overnight. After washing the plate with wash buffer (PBS with 0.05% Tween) twice, 200 µL of 4%

(w/v) bovine serum albumin in PBS were added as block buffer and incubated for 1 h, RT. After washing (4×), 50 µL of the 1:16000 diluted plasma sample were applied and incubated for 2 h, at RT. After 4 washing steps, 50 µL/well of a peroxidase-conjugated anti-human Fcgamma F(ab')2 fragment (1:20,000, Jackson ImmunoResearch, Dianova, Germany) were applied for detection and incubated for 1 h, RT followed by washing (4×). In all, 50 µL/well of TMB substrate was applied and incubated for 3.5 min followed by a reaction stop with 50 µL/well of 1 M H$_3$PO$_4$. Plates were measured with a Spark reader (TECAN GmbH, Crailsheim, Germany). Concentrations were calculated using the SparkControl magellan software 2.2 (TECAN GmbH). To determine HA recombinant-specific IgG levels, Immulon 4 HBX plates (ThermoFisher, Massachusetts, USA) were coated with 50 µL/well of recombinant HA3 protein (2 µg/mL) and incubated at 4 °C overnight. The recombinant HA proteins of the strains Influenza A/Hong Kong/4801/2014, A/Kanas/14/2017, and A/Hong Kong/2671/2019 were used. The HA protein of the strain A/Singapore/INFIMH-16-0019/2016 (H3N2) from the vaccine in the season 2018/2019 was not available. Instead, HA of A/Hong Kong/4801/2014 was used for analysis, which shows 99% amino acid sequence similarity to A/Singapore/INFIMH-16-0019/2016 (H3N2). After washing the plate with wash buffer (PBS with 0.1% Tween, TPBS) three times, 100 µL of TPBS containing 3% non-fat dry milk (AmericanBio, Canton, MA, USA) were added as block buffer and incubated for 1 h, RT. After washing (4×), 50 µL of the 1:300 diluted plasma sample (with 1%-milk TPBS) were applied and incubated for 2 h, at RT. After 3 washing steps, 100 µL/well of an anti-human IgG Fab-specific horseradish peroxidase-conjugated antibody (Sigma-Aldrich, St. Louis, Missouri, USA, 1:3000 diluted with 1%-milk TPBS) were applied for detection and incubated for 1 h, RT followed by washing (3×). In all, 100 µL/well of SigmaFast O-phenylenediamine dihydrochloride substrate was applied and incubated at RT followed by a reaction stop with 50 µL/well of 3 M HCl. Plates were measured with a Synergy H1 hybrid multimode microplate reader (BioTek, Winooski, VT, USA) at 490 nm.

**Clustering and heatmap.** FlowSOM clustering was performed based on lineage and activation markers: CXCR5, PD-1, CXCR3, ICOS; CD38, CD127, Ki67, TOX; T-bet, TCF-1 (xdim = 2, ydim = 2, random seed 7176). Cluster frequencies were exported from the Omiq software. The cluster composition was then visualized using the GraphPad Prism software. Mean signal intensity of marker expression in FlowSOM clusters was determined and visualized with the Omiq software (Omiq, Inc., 2020).

**tSNE analysis.** Phenotypic analysis of CD4 T cell responses was performed longitudinally in early and late responders. All influenza-specific CD4 T cell responses were grouped according to their self-reported vaccination history (Fig. 1e). The samples were analyzed to visualize high-dimensional phenotypes on a two-dimensional map using tSNE, as previously described[36,64]. Briefly, live singlet non-naive influenza-specific CD4 T cells were included in the analysis. Gating strategies are shown in Supplementary Fig. 11. Two-dimensional tSNE representation was calculated (random seed 5229, max iterations 1000, perplexity 30, and theta 0.5) using the single-cell expression information from 10 antibody costainings (CXCR5, PD-1, CXCR3, ICOS; CD38, CD127, Ki67, TOX; T-bet, TCF-1) and analyzed using the Omiq software (Omiq, Inc., 2020).

**Statistics and reproducibility.** Statistical tests and graphical visualization were performed using the GraphPad 8 software (GraphPad Prism Software Inc., San Diego, CA, USA). Due to the small sample size, non-parametric tests were used. Paired data were analyzed with two-tailed Wilcoxon matched-pairs signed rank test. Unpaired data were analyzed with two-tailed Mann–Whitney U-test. In case of multiple comparisons, Bonferroni correction was applied. Adjusted levels for statistical significance are stated in the figure legends. Correlation analyses were performed by two-tailed Spearman correlation. A p value < 0.05 was considered statistically significant with the following significance levels: *p < 0.05; **p < 0.01; ***p < 0.001; ****p < 0.0001. ROC analysis was performed with 95% confidence interval and Wilson/Brown method. For visualization of summarized data, heatmaps were generated with the GraphPad 8 software. Heatmap analysis with unbiased clustering was done with R Studio (https://www.r-project.org) using the ComplexHeatmap package[65]. Spider plots were generated with Microsoft Excel 2010. Human samples are very limited, so it is not possible to perform experiments repeatedly.

**Reporting summary.** Further information on research design is available in the Nature Research Reporting Summary linked to this article.

## Data availability

Additional supporting data are available from the corresponding authors upon reasonable request (response within 2 weeks). All requests for raw and analyzed data and materials will be reviewed by the corresponding authors to verify whether the request is subject to any intellectual property or confidentiality obligations. Donor-related data not included in the paper were generated as a part of clinical examination. Any data and materials that can be shared will be released via a Material Transfer Agreement. Source data are provided with this paper.

## Code availability

R code to reproduce the analyses of multiparametric flow cytometric data is available online: https://github.com/teximmed2-fr/flu-specific-CD4-T-cells. The R code is citable with the following: https://doi.org/10.5281/zenodo.554109[66].

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

## Acknowledgements

We thank all healthy volunteers who contributed to this study; Dr. Sebastian Merker and Dr. Ulrike Burk for excellent administrative assistance; and Marissa Russ, Franziska Engels, Sebastian Zehe, and Oezlem Sogukpinar for excellent technical assistance. This work was supported by internal funds from the University Hospital Freiburg and grants from the Deutsche Forschungsgemeinschaft (DFG, German Research Foundation)—project 272983813—TP01 to R.T., TP02 to C.N.-H., TP04 to T.B., TP20 to M.H., and TP21 to B.B.

## Author contributions

Study concept and design: T.B., acquisition of data: K.W., M. Smits, S.S., analysis and interpretation of data: K.W., Z.K., S.K., F.K., T.B., drafting of the manuscript: K.W., Z.K., T.B.; critical revision of the manuscript for important intellectual content: M.H., C.N.-H., B.B., R.T.; statistical analysis: K.W., Z.K., obtained funding: C.N.-H., B.B., F.K., M.H., R.T., T.B.; technical or material support: F.K., M. Schwemmle; study supervision: T.B., R.T.

## Funding

## Competing interests

The authors declare no competing interests.
