## [Peer Review File · Nature Communications]

Pre-existing immunity and vaccine history determine hemagglutinin-specific CD4 T cell and IgG response following seasonal influenza vaccinationREVIEWER COMMENTS

Reviewer #1 (Remarks to the Author):

This manuscript describes highly sophisticated studies to dissect the nature of human CD4 T cells responses to influenza vaccination, using HLA-DR tetramer staining. This is an important subject to evaluate. The peptide specificity tracked is one that has been conserved in H3 and is compared to a separate peptide segment that has drifted. This is a nice experimental design. Cell surface proteins and transcription factors that distinguish functional subsets of CD4 T cells are implemented and the data are analyzed, for the most part, in an unsupervised fashion, all excellent features of the manuscript. The quality of the primary measurements of tetramer positive cells shown in supplemental data are impressive and the subsequent approaches appropriate for the questions addressed. Despite these strengths, the data, as presented, is difficult to interpret rigorously.

The representations of the data that emerges from these interesting studies are confusing however, most substantially in the manner that subjects and patterns are tracked and grouped. These bias the vast majority of the data presented.

1. Figure 1a, from which much of the resulting analyses are derived and the authors description is confusing. First, it appears that there are at least 4 patterns of the responses, rather than the 2 described by the authors. Although it is very difficult to track the subjects with this style of presentation, it seems that, based on gains in tetramer frequency over time, there are at least 5 non-responders-subjects who have no measurable and statistically significant gain in CD4 T cell frequencies post vaccination. This is in keeping with the findings of others. There appear to be 3-4 subjects who exhibit an early expansion detected at days 2-4 post vaccination, relative to their baseline frequencies of HA-specific CD4 T cells. There are at least 2 subjects who exhibit bimodal response kinetics and 2-4 late responders. The authors "lumping" these subjects into 2 groups "early" and "late", each (based on data presented in many of the subsequent figures into 9 and perhaps an equal number of the second category. The in order to evaluate these groups, the authors need to clearly measure expansion of the H3-specific cells quantified with the tetramer (perhaps as both "fold change" and the difference between BL and the day measured) at each time point, and then clearly and rigorously describe and justify the criteria for assignment into, minimally non-responders, early and late responders. All of the remaining data should be analyzed with these criteria and groupings and should focus on those vaccine recipients that exhibited a statistically significant response. They may also wish to better understand features of the CD4 T cells in the "non-responders" but this should be as a separate group. It is clear that there are multiple populations of CD4 T cells that are elicited by vaccination, it is just not clear that these track with the "early" and "late" kinetics and yet all of the data are presented in this framework. This makes interpretation of these data very problematic.

2. Related to the point above, in Figure 2f, from the Figure legend the authors have tracked the fold change over baseline through the course of the response. Without knowing which subjects are being tracked here, there is considerable overlap at each time point among the "early" and "late" responders. The same is seen with ICOS expression quantified in Figure 3a and Fig 3f. The authors should provide more information on the response patterns for each piece of the data measured by simply giving each subject a unique symbol distinguished by shape and color so that the reader can track each response and each phenotype over time, rather than in aggregate, which can of course be represented in parallel.

3. The criteria for previously vaccinated subjects was not clear. Were these self reported vaccine histories or vaccine records accessed by the study investigators? The authors should clarify.

4. In general, the authors do not adequately cite the considerable literature on the relationship between CD4 T cell responses and antibody responses to vaccination or effects of repeated vaccination on antibody and CD4 T cell responses, nor do they cite the models that have been put forth to explain these effects or the impact of pre-existing HA specific antibodies on subsequent antibody and CD4 T cell responses. There is also considerable data on the relationship between CD4 T cell expansion or Tfh expansion and antibody responses. They thereby unfairly represent many of their findings as novel. This literature should be cited. These authors do have novel findings, due to their impressive approaches, and these should be emphasized.

Reviewer #2 (Remarks to the Author):

The authors investigated the dynamics of CD4 T cells after influenza vaccination against a conserved epitope within the HA of the H3N2 strain. They were able to detect distinct dynamics in T cells elicited in subjects who did or didn't receive a seasonal influenza vaccine in the prior season, adding a mechanistic component to a previously described phenomenon. The authors further generated data that might indicate that higher levels of pre-existing antibodies negatively affect CD4 T cell activation. Overall the manuscript is of interest to the field of influenza vaccinology, but some points of concern should be addressed by the authors.

- The epitope within H3 was selected based on its conservation throughout multiple seasons. However, how representative is the response against this specific epitope for the overall CD4 response? Immunodominance for CD4 epitopes has been previously described and might also play a role in the different observed responses in this study.
- Inactivated influenza vaccines also contain substantial amounts of NP, with a number of conserved T cell epitopes. Could the authors discuss the potential of cross-presentation? Would it be possible that NP-specific CD4 T cells could activate HA-specific B cells?
- The serology performed in the study is inadequate. The use of ELISA is appropriate, but recombinant H3 protein should be used as antigen instead of the whole vaccine preparation. Since the epitope is restricted to H3, the strain-specific response would be the most relevant. Hemagglutination inhibition should be performed at least against H3N2. Since the HAI only measures a subset of antibodies it would be of interest if the observed correlations are still maintained in that assay.
- Line 83/97: Please specify that the epitope is found in H3.
- In figure 4 it seems like some of the early responders also did not show strong antibody responses. Please discuss the potential reasons.
- Lines 223-225: Please clarify that fold induction and not baseline titers were lowest in the most recently vaccinated group.
- Line 318: Is it possible that in addition to insufficient uptake, heavily opsonized antigen might still get taken up but processed in a way that does not result in effective presentation on MHCs?

Reviewer #3 (Remarks to the Author):

This is an interesting manuscript and continues investigation into the role of circulating T follicular helper cells (cTfh) in inducing a response to influenza vaccine which has been undertaken by several groups. Given the importance of T cell help in the humoral response to immunisation, data from experiments exploring this in humans are of direct relevance to progress in vaccinology.

There are several areas to which the authors may wish to pay attention, including the definition of the CD4+ T cell subsets under investigation, which are usually referred to as cTfh (found in the circulation) rather than Tfh (found in secondary lymphoid tissue). The ontogeny, function, and relationship of these two cell subsets remains incompletely defined.

The reasons for studying certain markers and transcription factors should be outlined at the beginning and the text needs to place the research question and the findings more securely in the context of what is already known in the field. This will also aid a more thorough interpretation of the data.

Introduction

Lines 72 to 76 introducing Tfh. Please distinguish between Tfh which are found in secondary lymphoid tissue and cTfh which are found in the circulation amongst bulk circulating CD4.

Clarify which subset has been measured in the study. If secondary lymphoid tissue was not tested, please be explicit about what tissue was analysed.

Please refer more completely to other previous findings in the field.

Results

Line 104-5: how many individuals were enrolled?

Did these individuals receive influenza vaccine every year, or only some of the years referred to in the text? How was the history of receiving these vaccinations elicited?

The design of the study needs clear explanation. Which vaccine (s) was being used to immunise participants? What was the timeline of blood sampling?

111-112 "CD4 T cells specific for a mutated epitope are not responsive to the vaccination" The subject and object in this sentence are unclear please consider re-phrasing.

124 "strong up-regulation of Tfh and activation markers". What is meant by this; are cTfh being induced by vaccination with influenza vaccine? The text should distinguish between findings based on MFI and those based on frequency and how these are interpreted.

143 Define TCF-1. Why was this chosen for the study?

144-145 cluster 3 – what else defines this cluster apart from T-bet?

158-159 "Importantly, vaccination history of the vaccinees was closely associated with the "early" and "late" ..." given the importance of this finding, vaccination history and how it was elicited need to be pre-defined and described in the manuscript.

201-2 "validating TOX as a transcriptional regulator in antigen-specific Tfh cells in humans." This assertion is incompletely supported by the data which are correlative. The role of TOX in cTfh development may need to be confirmed experimentally in human cells in future work. A review of what is known of transcriptional regulators of Tfh differentiation should also be included in the introduction. The reference (number 22) is to Tox2 as a target of Bcl6; the relationship between TOX and Tox2 should be made clear in the manuscript.

222 "Interestingly, BL vaccine-specific antibodies were highest". This finding is not unexpected. Please consider re-phrasing.

Discussion

More thought needs to go into the discussion around the function of cTfh and what these data show that adds to previous work in the field. For example, the frequency of CD38+ cTfh was higher post immunisation with influenza vaccine in those who had not been immunised in the preceding three years compared with those who had (M Cole et al. Responses to Quadrivalent Influenza Vaccine Reveal Distinct Circulating CD4+CXCR5+ T Cell Subsets in Men Living with HIV | Scientific Reports (nature.com))

295 "In addition, we were able to confirm the Tfh lineage of CXCR5+PD-1+ CD4 T cells on the transcriptional level." This assertion overestimates the impact of the data; please also see comment in the results.

298 The fact that Bcl-6 is not detectable in cTfh may reflect differences in Tfh and cTfh.

307 There was a difference in phenotype at baseline (CD127 expression). There needs to be a more thorough discussion of the importance and relevance of CD127 in T cell function and long-term maintenance of T cell subsets.

There is discussion of "negative interference" but no consideration of immune regulation. The immune response is necessarily finite, and it may not be an advantage to generate a strong antibody response in the context of already high titres of circulating antibody; it is expected that this response would be

regulated.

329-335. This is speculation. Where are the data that mRNA vaccines are more stable inducers of T cell activation than vaccines of other design, particularly adenoviral vectored vaccines?

Methods

These need a timeline and clearer description of the groups and vaccination history.

Tables

Where are the demographics of participants – age, sex, ethnicity etc?

Figures

Fig 2 What is the gating strategy for the t-SNE?

Although divided into three clusters, the data appear more like four clusters

Cluster 2 looks like two clusters based on CXCR5 expression: one that is CXCR5 lo and one that is CXCR5 mid to hi. CXCR5 hi cells in this cluster express high levels of cTfh activation markers. There also appears to be a cluster of CXCR5+ cells that appear in cluster three.

Fig 4 Difference in fold change between early and late is driven by four individuals – these findings are overstated in the text. Is there anything different about these individuals that can be detected clinically or immunologically? This affects every panel of this figure and therefore needs addressing.

Fig 5 Is the expression of CD127 highest in those driving the changes in Fig 4? Are there the same four individuals in panels e and f as in Fig 4?

Supplementary Tables and Figures

Supplementary Fig 2e It appears there are at least two individuals with ICOS+CD38++ HA 306-318-specific CD4 T cells at Day 7 and beyond; please comment on this in the text. Where are the data demonstrating no change in the total frequency of HA 306-318-specific CD4 T cells rather than the frequency of activated cells?

Fig4c Are there baseline data for 2-NBDG uptake?

Minor comments

The words interestingly and importantly are used repeatedly throughout. Please consider re-phrasing these sentences and allow the reader to decide on these aspects.

Check for grammatical and spelling errors

We sincerely thank the editorial team and the reviewers for spending their precious time with our manuscript and for providing their valuable comments. Following the reviewers' suggestions and the editorial guidance, we have performed additional experiments and revised the manuscript extensively and have responded to the reviewers' comments point-by-point.

Reviewer comments:

Reviewer #1 (Remarks to the Author):

This manuscript describes highly sophisticated studies to dissect the nature of human CD4 T cells responses to influenza vaccination, using HLA-DR tetramer staining. This is an important subject to evaluate. The peptide specificity tracked is one that is has been conserved in H3 and is compared to a separate peptide segment that has drifted. This is a nice experimental design. Cell surface proteins and transcription factors that distinguish functional subsets of CD4 T cells are implemented and the data are analyzed, for the most part, in an unsupervised fashion, all excellent features of the manuscript. The quality of the primary measurements of tetramer positive cells shown in supplemental data are impressive and the subsequent approaches appropriate for the questions addressed. Despite these strengths, the data, as presented, is difficult to interpret rigorously.

Response: We thank the reviewer for the positive assessment of our study and for highlighting the methodological strengths.

The representations of the data that emerges from these interesting studies are confusing however, most substantially in the manner that subjects and patterns are tracked and grouped. These bias the vast majority of the data presented.

1. Figure 1a, from which much of the resulting analyses are derived and the authors description is confusing. First, it appears that there are at least 4 patterns of the responses, rather than the 2 described by the authors. Although it is very difficult to track the subjects with this style of presentation, it seems that, based on gains in tetramer frequency over time, there are at least 5 non-responders-subjects who have no measurable and statistically significant gain in CD4 T cell frequencies post vaccination. This is in keeping with the findings of others. There appear to be 3-4 subjects who exhibit an early expansion detected at days 2-4 post vaccination, relative to their baseline frequencies of HA-specific CD4 T cells. There are at least 2 subjects who exhibit bimodal response kinetics and 2-4 late responders. The authors "lumping" these subjects into 2 groups "early" and "late", each (based on data presented in many of the subsequent figures into 9 and perhaps an equal number of the second category. The in order to evaluate these groups, the authors need to clearly measure expansion of the H3-specific cells quantified with the tetramer (perhaps as both "fold change" and the difference between BL and the day measured) at each time point, and then clearly and rigorously describe and justify the criteria for assignment into, minimally non-responders, early and late

responders. All of the remaining data should be analyzed with these criteria and groupings and should focus on those vaccine recipients that exhibited a statistically significant response. They may also wish to better understand features of the CD4 T cells in the "non-responders" but this should be as a separate group. It is clear that there are multiple populations of CD4 T cells that are elicited by vaccination, it is just not clear that these track with the "early" and "late" kinetics and yet all of the data are presented in this framework. This makes interpretation of these data very problematic.

Response: We appreciate the reviewer's comment regarding the different response patterns and the inherent difficulties to group these response without introducing a selection bias. First of all, we apologize for not presenting the data in a clearer fashion that might have created some confusion: We very much agree with the reviewer that several different expansion patterns can be observed among the vaccinees, when looking at the frequencies of HA₁₁₈₋₁₃₂-specific CD4 T cells and their changes over time. However, the separation of groups (early vs. late) in the initial submission was not made based on the changes in frequency, but rather changes in the activation patterns of HA₁₁₈₋₁₃₂-specific CD4 T cells and this was done by unsupervised clustering of a heatmap containing the different surface markers in order to avoid selection bias (beyond the "selection of the relevant markers"). In addition, we agree that by purely looking at frequencies, the impression may arise that some individuals are "non-responders" (e.g. #2-2019, #5-2016 and #12-2020). However, in contrast to their low proliferative response, HA₁₁₈₋₁₃₂-specific CD4 T cells from these donors were strongly activated on day 4 as indicated by co-expression of ICOS and CD38, clearly demonstrating a response to the vaccine on the influenza-specific CD4 level (Fig. 2). As the reviewer correctly points out, the expansion patterns of HA₁₁₈₋₁₃₂-specific CD4 T cells among the different time points are very heterogeneous, making it almost impossible to identify recurring patterns that could subsequently be used throughout the manuscript. In addition, the fold change in frequency from baseline to days 4, 7 or 14 did not correlate with IgG induction (Fig. 5d, Fig.6c). Therefore, we have not grouped the vaccinees according to their changes in tetramer frequency as this would surely have introduced some form of selection bias and would not have resulted in meaningful correlations with the most relevant endpoint. However, we have specifically addressed this aspect in the revised manuscript in order to avoid confusion:

Results:

Line 182f: In contrast, the FC of the HA₁₁₈₋₁₃₂-specific CD4 T cell frequency at day 4 showed no correlation with the FC of HA-specific or vaccine-specific antibodies (Fig. 5d and Supplementary Fig. 6d), indicating that phenotypic changes more specifically associate with antibody induction than the FC of the HA₁₁₈₋₁₃₂-specific CD4 T cell frequency.

Line 200f: None of the baseline parameters correlated with the FC in HA₁₁₈₋₁₃₂-specific CD4 T cell frequency from baseline to day 4 (Fig. 6c).

Nevertheless, the reviewer point regarding the presentation of the data is very well taken. Therefore, we have made substantial changes to the figure design and the grouping of the individuals and have now sorted the vaccinees color-coded by vaccine history in order to improve the readability of the manuscript.

2. Related to the point above, in Figure 2f, from the Figure legend the authors have tracked the fold change over baseline through the course of the response. Without knowing which

subjects are being tracked here, there is considerable overlap at each time point among the “early” and “late” responders. The same is seen with ICOS expression quantified in Figure 3a and Fig 3f. The authors should provide more information on the response patterns for each piece of the data measured by simply giving each subject a unique symbol distinguished by shape and color so that the reader can track each response and each phenotype over time, rather than in aggregate, which can of course be represented in parallel.

Response: We thank the reviewer for this suggestion to improve the presentation of the data. In the revised version, we have now assigned an individual symbol to all donors/year. The color constellation is assigned to vaccination history (blue = vaccinated in the previous year, turquoise = last vaccine received > 1 year ago [range 2-10 years], red = vaccine-naive). We have changed this presentation method for the entire manuscript so that the phenotypic characterization (Expression of memory marker CD127, proliferation marker Ki67, activation marker ICOS and CD38 and Tfh associated marker CXCR5+PD-1 on HA₁₁₈₋₁₃₂ specific CD4 T cells) of the data is clearer and more comprehensible.

3. The criteria for previously vaccinated subjects was not clear. Were these self reported vaccine histories or vaccine records accessed by the study investigators? The authors should clarify.

Response: We apologize for the missing information. Study participants self-reported their vaccination history and most of them provided vaccination records. This information is added in the revised manuscript.

Line 306: *The individual vaccination responses are grouped by the donors' self-reported vaccination history.*

4. In general, the authors do not adequately cite the considerable literature on the relationship between CD4 T cell responses and antibody responses to vaccination or effects of repeated vaccination on antibody and CD4 T cell responses, nor do they cite the models that have been put forth to explain these effects or the impact of pre-existing HA specific antibodies on subsequent antibody and CD4 T cell responses. There is also considerable data on the relationship between CD4 T cell expansion or Tfh expansion and antibody responses. They thereby unfairly represent many of their findings as novel. This literature should be cited. These authors do have novel findings, due to their impressive approaches, and these should be emphasized.

Response: We agree with the reviewer's criticism that several relevant studies have not been adequately referenced in the introduction and/or discussion and apologize for this shortcoming. We have now revised the introduction and discussion and have cited the following studies that were not included in the previous submission:

Parts of the revised introduction (numbers of references differ in the manuscript):

Lines 82ff: *Previous studies analyzing bulk cTfh cells after influenza vaccination identified a peak of activation on day 7 characterized by CD38, ICOS and CXCR3 expression(1-4). Similar to observations on antibody titers after repeat vaccination, this was more strongly pronounced in individuals without self-reported influenza vaccination in the preceding years(5). In other*

studies, activation-induced marker or cytokine expression after antigen-stimulation confirmed the presence of antigen-specific CD4 T cells within this population(6, 7) and an increase of this population correlated with the amount (1, 3, 7, 8) and the avidity(2) of the flu-specific antibodies, suggesting a mechanistic connection between cTfh responses and vaccine-elicited antibodies(4, 5, 9). However, few studies have aligned the MHC class II epitope-sequences with the current vaccine strain or have phenotypically analyzed influenza-specific CD4 T cell responses at baseline and earlier than day 7 after vaccination(1, 10).

Parts of the revised discussion:

Lines: 216ff: Importantly, several studies have shown a direct relationship between the frequency of activated cTfh cells following vaccination and influenza vaccine-induced antibody responses(4, 5, 9, 11). However, while antibody responses are typically analyzed in an antigen-specific fashion, activation or expansion patterns of their T cell counterparts have mostly been measured on bulk populations(1-5, 9, 11), hindering conclusions on protein-specific T cell-B cell interactions.

Lines 230ff: Indeed, it has previously been shown that repeat influenza vaccination reduces the antibody induction and impairs the affinity maturation of influenza-specific antibodies(12-17). This observation has also been linked to the activation of bulk cTfh cells 7 days after vaccination that has been shown to be lower in those individuals that had received the vaccine in the previous years compared to those who did not(5).

Lines 239f: Similar to previous observations on the bulk level(5, 9), the individual vaccine history was closely associated with the presence of the “early” activation pattern and most vaccinees displaying strong CD4 T cell activation had not received a vaccine in the previous year (Fig. 3a).

Lines 267ff: One possible explanation that has been brought forth to explain ineffective seasonal vaccination is the concept of the original antigenic sin (OAS) which describes the phenomenon that antibodies targeting epitopes of the first strain an individual has been exposed to are predominantly boosted after vaccination at the expenses of antibodies targeting novel epitopes(18, 19). However, this concept may not apply in our study focusing on T cells targeting a conserved epitope that are differentially activated in repeated vaccinations even within the same individuals (Supplementary Fig. 4). Alternatively, pre-existing antibodies have been suggested to mask relevant epitopes which could subsequently prevent the stimulation of specific B cells(20, 21). Thus, it is equally conceivable that such a mechanism might result in insufficient uptake of antigens by antigen presenting cells (APCs) and subsequently reduced presentation via MHC class II to influenza-specific CD4 T cells. Furthermore, IgG-opsonized antigens (immune complexes) influence antigen availability and processing by binding to Fcγ-Receptors on APCs and B cells. Thereby, the different route of antigen-uptake is a key determinant for further antigen processing and presentation(22-24). “Negative interference” of pre-existing antibodies with antigen availability uptake and presentation, might also be involved in the context of the “antigenic distance hypothesis” (17, 25). Indeed, this hypothesis was developed based on the observation that in certain contexts, repeat vaccinations suffer from limited efficacy and postulates that a small antigenic distance between two vaccine strains results in limited vaccine efficacy, especially when the season’s epidemic strain is antigenically more distant (26). While the clinical relevance of this theory with respect to vaccine efficacy and translation into protective efficacy remains to be proven, the concept of “negative interference” could nevertheless explain the weak HA₁₁₈₋₁₃₂-specific CD4 T cell activation and

subsequent impaired antibody induction after repeat vaccination. In addition, the inverse correlation between preexisting antibodies and the antibody induction may suggest that a certain target level of antibodies specific for a given antigen is centrally regulated(4, 5, 27). This assumption would be supported by our observation that antibody levels were similar between groups at day 28 (Fig. 5c), suggesting that an immune-rheostat strives for a certain target level and prevents immune activation in response to the vaccine if the target level is already reached. In contrast to this hypothesis, it has recently been shown that individuals receiving SARS-CoV-2 vaccination after natural infection show stronger antibody responses than those with either natural infection or vaccination alone(28-31). The sequence of natural infection followed by vaccination results in a state termed "hybrid immunity" and the higher antibody titers observed in these individuals has been attributed to the idea that the host perceives repeat antigenic exposure as an increased threat and subsequently strives for higher antibody titers(32).

Reviewer #2 (Remarks to the Author):

The authors investigated the dynamics of CD4 T cells after influenza vaccination against a conserved epitope within the HA of the H3N2 strain. They were able to detect distinct dynamics in T cells elicited in subjects who did or didn't receive a seasonal influenza vaccine in the prior season, adding a mechanistic component to a previously described phenomenon. The authors further generated data that might indicate that higher levels of pre-existing antibodies negatively affect CD4 T cell activation. Overall the manuscript is of interest to the field of influenza vaccinology, but some points of concern should be addressed by the authors.

Response: We thank the reviewer for the positive assessment of our study and for highlighting the potential biological relevance.

1. The epitope within H3 was selected based on its conservation throughout multiple seasons. However, how representative is the response against this specific epitope for the overall CD4 response? Immunodominance for CD4 epitopes has been previously described and might also play a role in the different observed responses in this study.

Response: The reviewer brings up a very important point. Indeed, the role of immunodominance for CD4 T cell epitopes is essential to demonstrate that the response of this specific epitope is representative for the overall CD4 response. The HA₁₁₃₋₁₃₂ epitope was first described by Uchtenhagen et al. in 2016. Using an overlapping 20-mer HA peptide experiment, Uchtenhagen et al. were able to identify 4 epitopes that showed the strongest IFN- γ response (10). In this context, HA₁₁₃₋₁₃₂ was identified and restricted to DRB1*0401.

We used *in silico* MHC class II epitope prediction for DRB1*0101 to identify the best 15mer HA binder (HA₁₁₈₋₁₃₂). An internal experiment with EBV immortalized B cells as antigen-presenting cells confirmed the restriction to DRB1*0101 (see figure below). Even with a very low concentration of 0.0625 μ M HA₁₁₈₋₁₃₂, we could detect an IFN- γ response indicating a high affinity to bind to the HLA type DRB1*0101. Thus, also linking this epitope to DRB1*0101. Regarding immunodominance, we were able to detect HA₁₁₈₋₁₃₂ specific CD4 T cells in all 12 tested donors. Thus, we believe that the data by Uchtenhagen et al. and the 100% detection

rate in all donors in our experiments on a different HLA background establish HA₁₁₈₋₁₃₂ as an immunodominant epitope.

2. Inactivated influenza vaccines also contain substantial amounts of NP, with a number of conserved T cell epitopes. Could the authors discuss the potential of cross-presentation? Would it be possible that NP-specific CD4 T cells could activate HA-specific B cells?

Response: We are not entirely sure that we understand the question correct. Cross-presentation is typically defined by the presentation of epitopes from exogenous antigens on MHC I molecules on antigen presenting cells (APC). And although this concept might be relevant in the context of influenza vaccination with split and subunit vaccines, which were used in our study, where the antigen is exogenous despite it's viral origin and could be cross-presented to HA-specific CD8 T cells by dendritic cells (DCs) and also B cells (33, 34), we believe that the reviewer does not refer to this concept but would rather interpret the first question on cross-presentation in the context of the second question on the possibility of NP-specific CD4 T cells activating HA-specific B cells.

We had focused specifically on HA-specific CD4 T cell immunity, mostly because the immune response after influenza vaccination predominantly targets HA, and only to a lesser extent NP (35). We therefore cannot comment on how well NP-specific CD4 T cells are activated after vaccination. If we hypothesize they would be activated and would come in close proximity to B cells specific for a different protein, they still should not be able to directly activate the HA-specific B cell, unless they are activated by another NP-specific B cell. If an HA-specific B cell would present epitopes of NP and HA at the same time, then NP- as well as HA-specific CD4 T cells that recognize the presented epitopes could activate the B cell by cognate interaction. The simultaneous presentation of H3 and NP after vaccination would be possible if the HA and NP antigens are linked and therefore taken up together, when the BCR of the HA-specific B cell binds to HA. Then the whole molecule would be processed and resulting epitopes presented accordingly (reviewed in (36, 37). Since HA and NP are not linked together as one molecule in the vaccine, a simultaneous presentation of HA and NP on the B cell and therefore also the activation of the HA-specific B cell by an NP-specific CD4 T cell is rather unlikely. However, contact-dependent B cell activation also occurs independently of antigen recognition, which has been shown previously *in vitro* (38).

As we are not entirely sure we understood the essence of the reviewer's question, we have not added this discussion to the manuscript, but would be happy to do so, in case the reviewer believes this to be important.

3. The serology performed in the study is inadequate. The use of ELISA is appropriate, but recombinant H3 protein should be used as antigen instead of the whole vaccine preparation. Since the epitope is restricted to H3, the strain-specific response would be the most relevant. Since the HAI only measures a subset of antibodies it would be of interest if the observed correlations are still maintained in that assay.

Response: We fully understand the concern of the reviewer that the antibody detection could be more aligned with CD4 T cell specificity we analyzed in our study. Thus, we teamed up with Florian Krammers lab in order to measure antibody responses against the recombinant H3 protein from each vaccine strain and added the results to the revised manuscript (Fig. 5 and 6). Since the H3 protein from A/Singapore/INFIMH-16-0019/2016 was not available, the H3 protein from A/Hong Kong/4801/2014 with a 99% sequence homology was used for samples from the 2018 vaccine season. However, testing all samples against all three H3 proteins revealed almost identical results with the exception of the 2019 samples that showed lower FC when tested against the H3 protein from A/Hong Kong/4801/2014 compared to the H3 protein from A/Kansas/14/2017 (which corresponds to the 2019 vaccine strain and has a 98% homology to the HongKong strain). These analyses largely confirmed the results with the entire vaccine. We therefore chose to also present these data in the supplement.

4. Line 83/97: Please specify that the epitope is found in H3.

Response: We have added this information to the revised manuscript.

Lines 80f: [...], we focused on two HLA-DRB1*01:01- MHC class II-restricted HA3-derived epitopes (HA₁₁₈₋₁₃₂ VPDYASLRSLVASSG and HA₃₀₆₋₃₁₈ PKYVKQNTLKLAT) in 12 individuals (Table 1).

5. In figure 4 it seems like some of the early responders also did not show strong antibody responses. Please discuss the potential reasons

Response: We appreciate the reviewer's comment and have addressed this aspect in the revised version of the manuscript. As discussed in response to reviewer #1, we have now assigned the vaccinees to groups based on their vaccination history. With regards to the antibody response against the entire vaccine, shown in the initial submission, the vaccination history is a strong separator and those early responders not showing strong antibody responses against the entire vaccine belong to the recently vaccinated group. However, in response to the earlier comment by this reviewer, we have performed new serologies, specifically analyzing H3-specific antibody responses. These new data largely confirm the results obtained by analyzing antibody responses against the entire vaccine but more closely associate the T cell activation patterns with the antibody response. Collectively, our data now demonstrate that an early activation of HA₁₁₈₋₁₃₂-specific CD4 T cell responses is predictive of an increase in H3-specific antibodies after influenza vaccination, more so than the vaccination history alone (Fig. 5b).

6. Lines 223-225: Please clarify that fold induction and not baseline titers were lowest in the most recently vaccinated group.

Response: We thank for this comment. Due to restructuring of the manuscript, this sentence has been deleted.

7. Line 318: Is it possible that in addition to insufficient uptake, heavily opsonized antigen might still get taken up but processed in a way that does not result in effective presentation on MHCs?

Response: The reviewer raises the interesting question how pre-existing antibodies influence an immune response besides the possibility of neutralization and insufficient uptake of the antigen.

Indeed, IgG-opsonized antigens (immune complexes) influence antigen availability and processing by binding to Fcγ-Receptors on APCs and B cells. Thereby, the different route of antigen-uptake (receptor-mediated endocytosis vs phagocytosis or micropinocytosis) is a key determinant (39-41). The uptake of immune complexes utilizes some distinct molecular mechanisms compared to those of phagocytosis of larger particles. Additionally, the different FcγRs vary in their affinity for IgG and use different intracellular trafficking pathways. By controlling the shuttling and processing of the immune complexes they are able to regulate antigen presentation (42). FcγR mediated signaling can have activating and inhibitory effects depending on the FcγR: monocyte-derived DCs and macrophages express high levels of activating FcγRs, whereas conventional and plasmacytoid DCs express the inhibitory FcγR (43). The Fc-mediated internalization of ICs has been associated with subsequent enhancement in cellular antigen presentation efficiency (44). Moreover, internalization of IgG immune complexes enhances cross-presentation in DCs compared to the fluid phase uptake of antigen (45). Furthermore, B cell activation is regulated by FcRs (46). The low affinity inhibitory receptor FcγRIIb is known as key regulator of B cells responses (47, 48) Collectively, the regulation of antigen processing and presentation is very complex and comprises activation and inhibitory signals. These are dependent on the cell population as well as the receptor availability and expression level. To evaluate if immune complexes influence antigen presentation after influenza vaccination of recently vaccinated individuals at least the FcγR repertoire and expression levels on APC populations and B cells would need to be quantified reveal details in this specific context. We have added this possible explanation that opsonized antigen might be processed and presented ineffectively on MHC to the revised version, which now reads (numbers of references differ in the manuscript):

Lines 271ff: *Alternatively, pre-existing antibodies have been suggested to mask relevant epitopes which could subsequently prevent the stimulation of specific B cells(20, 21). Thus, it is equally conceivable that such a mechanism might result in insufficient uptake of antigens by antigen presenting cells (APCs) and subsequently reduced presentation via MHC class II to influenza-specific CD4 T cells. Furthermore, IgG-opsonized antigens (immune complexes) influence antigen availability and processing by binding to Fcγ-Receptors on APCs and B cells. Thereby, the different route of antigen-uptake is a key determinant for further antigen processing and presentation (22-24).*

Reviewer #3 (Remarks to the Author):

This is an interesting manuscript and continues investigation into the role of circulating T follicular helper cells (cTfh) in inducing a response to influenza vaccine which has been undertaken by several groups. Given the importance of T cell help in the humoral response to immunisation, data from experiments exploring this in humans are of direct relevance to progress in vaccinology.

Response: We thank the reviewer for the positive assessment of our study and for highlighting the relevance to vaccine research.

There are several areas to which the authors may wish to pay attention, including the definition of the CD4+ T cell subsets under investigation, which are usually referred to as cTfh (found in the circulation) rather than Tfh (found in secondary lymphoid tissue). The ontogeny, function, and relationship of these two cell subsets remains incompletely defined. The reasons for studying certain markers and transcription factors should be outlined at the beginning and the text needs to place the research question and the findings more securely in the context of what is already known in the field. This will also aid a more thorough interpretation of the data.

Response: We thank the reviewer for bringing up these important aspects and apologize for being vague at times with regards to these aspects. We want to take this opportunity to clarify that our main focus was the analysis of influenza-specific CD4 T cells and to subsequently analyze changes within influenza-specific (in our case HA₁₁₈₋₁₃₂-specific) CD4 T cells in response to the vaccine. Thus, the pre-specified CD4 T cell subset under investigation was defined by its antigen-specificity, not its T helper lineage commitment. This is in contrast to many studies analyzing cTfh responses after vaccination, in which it often remains unclear which of these cells are actually influenza-specific since the pre-specified T cell population under investigation is defined by its lineage commitment and not its antigen-specificity. Thus, by longitudinally visualizing influenza-specific CD4 T cells that target a conserved epitope in combination with an analysis of lineage-defining surface markers and transcription factors, we are able to track the fate of these cells without introducing a selection bias by pre-defining a specific lineage that is analyzed irrespective of their antigen-specificity. However, in order to better introduce the specific lineages and markers, we have largely revised the introduction, as suggested by the reviewer and have also specifically addressed similar comments by the reviewer further below.

Introduction

Lines 72 to 76 introducing Tfh. Please distinguish between Tfh which are found in secondary lymphoid tissue and cTfh which are found in the circulation amongst bulk circulating CD4. Clarify which subset has been measured in the study. If secondary lymphoid tissue was not tested, please be explicit about what tissue was analysed.

Response: In response to this comment, we have introduced circulating Tfh cells and germinal center- / lymphoid-tissue-derived Tfh cells. As we have not taken biopsies or fine-needle aspirates longitudinally, we have also clarified that the analysis was done from blood samples (figure 1a). The corresponding sections in the manuscript now read as follows:

Introduction (numbers of references differ in the manuscript):

Lines 55ff: Among CD4 T helper cells, T follicular helper (Tfh) cells are key players in the generation of antibody responses as they provide the fundamental signals to B cells in the

germinal center (GC) reaction underlying affinity maturation. Tfh cells are characterized by expression of the CXC motif chemokine receptor 5 (CXCR5) and programmed death protein 1 (PD-1)(49). Since GC-derived Tfh cells are difficult to assess in humans, circulating Tfh (cTfh) cells are commonly used as surrogates as they share phenotypic, functional and clonal characteristics with lymphoid tissue derived GC-Tfh cells(50-55).

Results:

Lines 79f: In order to longitudinally analyze circulating influenza-specific CD4 T cells responding to the vaccine (Fig. 1a), [...].

Please refer more completely to other previous findings in the field.

Response: We agree with the reviewer's criticism that several relevant studies have not been adequately referenced in the introduction and/or discussion and apologize for this shortcoming. We have now revised the introduction and discussion and have cited the following studies that were not included in the previous submission:

Parts of the revised introduction:

Previous studies analyzing bulk cTfh cells after influenza vaccination identified a peak of activation on day 7 characterized by CD38, ICOS and CXCR3 expression(1-4). Similar to observations on antibody titers after repeat vaccination, this was more strongly pronounced in individuals without self-reported influenza vaccination in the preceding years(5). In other studies, activation-induced marker or cytokine expression after antigen-stimulation confirmed the presence of antigen-specific CD4 T cells within this population(6, 7) and an increase of this population correlated with the amount (1, 3, 7, 8) and the avidity(2) of the flu-specific antibodies, suggesting a mechanistic connection between cTfh responses and vaccine-elicited antibodies(4, 5, 9). However, few studies have aligned the MHC class II epitope-sequences with the current vaccine strain or have phenotypically analyzed influenza-specific CD4 T cell responses at baseline and earlier than day 7 after vaccination(1, 10).

Parts of the revised discussion:

Lines: 216ff: Importantly, several studies have shown a direct relationship between the frequency of activated cTfh cells following vaccination and influenza vaccine-induced antibody responses(4, 5, 9, 11). However, while antibody responses are typically analyzed in an antigen-specific fashion, activation or expansion patterns of their T cell counterparts have mostly been measured on bulk populations(1-5, 9, 11), hindering conclusions on protein-specific T cell-B cell interactions.

Lines 230ff: Indeed, it has previously been shown that repeat influenza vaccination reduces the antibody induction and impairs the affinity maturation of influenza-specific antibodies(12-17). This observation has also been linked to the activation of bulk cTfh cells 7 days after vaccination that has been shown to be lower in those individuals that had received the vaccine in the previous years compared to those who did not(5).

Lines 239f: Similar to previous observations on the bulk level(5, 9), the individual vaccine history was closely associated with the presence of the “early” activation pattern and most vaccinees displaying strong CD4 T cell activation had not received a vaccine in the previous year (Fig. 3a).

Lines 267ff: One possible explanation that has been brought forth to explain ineffective seasonal vaccination is the concept of the original antigenic sin (OAS) which describes the phenomenon that antibodies targeting epitopes of the first strain an individual has been exposed to are predominantly boosted after vaccination at the expenses of antibodies targeting novel epitopes(18, 19). However, this concept may not apply in our study focusing on T cells targeting a conserved epitope that are differentially activated in repeated vaccinations even within the same individuals (Supplementary Fig. 4). Alternatively, pre-existing antibodies have been suggested to mask relevant epitopes which could subsequently prevent the stimulation of specific B cells(20, 21). Thus, it is equally conceivable that such a mechanism might result in insufficient uptake of antigens by antigen presenting cells (APCs) and subsequently reduced presentation via MHC class II to influenza-specific CD4 T cells. Furthermore, IgG-opsonized antigens (immune complexes) influence antigen availability and processing by binding to Fcγ-Receptors on APCs and B cells. Thereby, the different route of antigen-uptake is a key determinant for further antigen processing and presentation(22-24). “Negative interference” of pre-existing antibodies with antigen availability uptake and presentation, might also be involved in the context of the “antigenic distance hypothesis” (17, 25). Indeed, this hypothesis was developed based on the observation that in certain contexts, repeat vaccinations suffer from limited efficacy and postulates that a small antigenic distance between two vaccine strains results in limited vaccine efficacy, especially when the season’s epidemic strain is antigenically more distant (26). While the clinical relevance of this theory with respect to vaccine efficacy and translation into protective efficacy remains to be proven, the concept of “negative interference” could nevertheless explain the weak HA₁₁₈₋₁₃₂-specific CD4 T cell activation and subsequent impaired antibody induction after repeat vaccination. In addition, the inverse correlation between preexisting antibodies and the antibody induction may suggest that a certain target level of antibodies specific for a given antigen is centrally regulated(4, 5, 27). This assumption would be supported by our observation that antibody levels were similar between groups at day 28 (Fig. 5c), suggesting that an immune-rheostat strives for a certain target level and prevents immune activation in response to the vaccine if the target level is already reached. In contrast to this hypothesis, it has recently been shown that individuals receiving SARS-CoV-2 vaccination after natural infection show stronger antibody responses than those with either natural infection or vaccination alone(28-31). The sequence of natural infection followed by vaccination results in a state termed “hybrid immunity” and the higher antibody titers observed in these individuals has been attributed to the idea that the host perceives repeat antigenic exposure as an increased threat and subsequently strives for higher antibody titers(32).

Results

Line 104-5: how many individuals were enrolled?

Response: We have mentioned the number of donors in the method part (see Design). To highlight the number in the results section, we added the number of donors also in the results section, have provided a table with the number of donors (Table 1).

Did these individuals receive influenza vaccine every year, or only some of the years referred to in the text? How was the history of receiving these vaccinations elicited?

Response: No, unfortunately not all participants could be studied in every year. In order for the reader to be able to track this information, we have assigned each participant a number, followed by the year in which he/she was vaccinated (i.e. #1-2016 shows data from the 2016 vaccination from participant #1). Study participants self-reported their vaccination history and most of them provided vaccination records. We added this information to the revised manuscript (see Table 1).

The design of the study needs clear explanation. Which vaccine (s) was being used to immunise participants? What was the timeline of blood sampling?

Response: We apologize that this information has not been provided more prominently in the initial submission. We have mentioned the design of our study and the timeline of blood sampling in the method part and have now included a sketch of the timeline for blood sampling to figure 1a. In addition, we had described the vaccines with which participants had been vaccinated in Table 1 and have now moved this table to the main document.

111-112 “CD4 T cells specific for a mutated epitope are not responsive to the vaccination” The subject and object in this sentence are unclear please consider re-phrasing.

Response: We have revised the sentence. It now reads:

Line 88f: *In contrast, frequencies of HA₃₀₆₋₃₁₈-specific CD4 T cells did not change, indicating that the sequence variations in the vaccine strain prevented the activation of this T cell specificity.*

124 “strong up-regulation of Tfh and activation markers”. What is meant by this; are cTfh being induced by vaccination with influenza vaccine? The text should distinguish between findings based on MFI and those based on frequency and how these are interpreted.

Response: We are not sure that we completely understand this comment. As discussed above, our approach was to analyze changes of the influenza-specific CD4 T cells in response to the vaccine and have therefore analyzed activation and lineage-defining markers longitudinally on influenza-HA₁₁₈₋₁₃₂-specific CD4 T cells. At baseline, none of the cells were activated and few displayed a Tfh phenotype. This would be expected as they have not been exposed to their cognate antigen in at least one year. In response to the vaccine (which includes the HA₁₁₈₋₁₃₂ epitope), Tfh and activation markers were differentially upregulated in the different individuals, demonstrating changes in the composition of the HA₁₁₈₋₁₃₂-specific CD4 T cell pool with more activated cells and more cells expressing markers indicative of Tfh cell differentiation. The question whether cTfh cells are being induced is certainly intriguing but cannot be answered by our results. We can safely state that markers that are used to identify

cTfh cells are increasingly expressed on HA₁₁₈₋₁₃₂-specific CD4 T cells after vaccination in a subset of individuals, but whether these cells are induced, transcriptionally reprogrammed or are even functionally capable of providing B cell help cannot be answered at this stage.

With regards to the comment on MFI and frequency: we are unsure which part of the manuscript the reviewer is referring to. We agree that MFI and frequency of expression of a certain marker are technically two different ways of analyzing a dataset. However, if a certain activation is increased within a population, both measures will provide very similar information and can therefore be interpreted in a similar manner. Regarding our manuscript, we could not identify a statement or an interpretation that would substantially change by using MFI instead of frequency and vice versa. However, we would be happy to re-analyze and discuss in detail if needed.

143 Define TCF-1. Why was this chosen for the study?

Response: We apologize for the missing introduction of the T cell factor 1 (TCF-1), a transcription factor, which is encoded by *TCF7*. TCF-1 is essential for the early steps in Tfh differentiation (56). In addition to its role in Tfh differentiation, TCF-1 is also relevant for memory T cell formation (57, 58). However, the expression patterns of TCF-1 during recall responses in human antigen-specific CD4 T cell responses are unknown, therefore, we included this important transcription factor in our study. The revised introduction now reads:

Lines 61ff: *The master transcription factor of Tfh cells, Bcl6(49) is not expressed on cTfh cells(52, 59) and the precise expression patterns of other transcription factors that have been associated with Tfh development, such as TOX and T cell factor 1 (TCF-1), remain to be identified(56, 60). Furthermore, they have not been established as lineage-defining markers for antigen-specific cTfh cells in humans as they are not exclusively active in the Tfh program(57, 60).*

144-145 cluster 3 – what else defines this cluster apart from T-bet?

Response: The defining markers that most prominently separate the individual clusters are displayed in the accompanying heatmap. In response to a later comment by this reviewer, we have modified the clustering analysis (please find our response further below).

158-159 “Importantly, vaccination history of the vaccinees was closely associated with the “early” and “late” ...” given the importance of this finding, vaccination history and how it was elicited need to be pre-defined and described in the manuscript.

Response: As mentioned above, study participants self-reported their vaccination history and most of them provided vaccination records. We added this information to the revised manuscript.

201-2 “validating TOX as a transcriptional regulator in antigen-specific Tfh cells in humans.” This assertion is incompletely supported by the data which are correlative. The role of TOX in cTfh development may need to be confirmed experimentally in human cells in future work. A review of what is known of transcriptional regulators of Tfh differentiation should also be

included in the introduction. The reference (number 22) is to Tox2 as a target of Bcl6; the relationship between TOX and Tox2 should be made clear in the manuscript.

Response: We thank the reviewer for this important comment and fully agree that more studies are required to confirm TOX as a transcription factor in human cTfh cells. Although TOX has never been analyzed during recall responses in human antigen-specific CD4 T cells with a cTfh phenotype, we have toned our statement which now reads:

Lines 160ff: *TOX expression was significantly higher in cTfh vs. non-cTfh HA₁₁₈₋₁₃₂-specific CD4 T cells early after vaccination (Fig. 4f and Supplementary Fig. 4c), suggesting that TOX might be involved in the emergence of cTfh cell responses within HA₁₁₈₋₁₃₂-specific CD4 T cells upon vaccination.*

In addition, we have revised the introduction to provide some context regarding the transcriptional regulators of Tfh differentiation (see comment above and lines 61ff in revised manuscript). However, transcriptional regulation of Tfh differentiation is not the main focus of our work, although we provide novel information on the expression levels of relevant transcription factors in human antigen-specific CD4 T cells and have therefore referenced a comprehensive review article summarizing this aspect in more depth.

Regarding the reference „*The Transcription Factor TOX2 Drives T Follicular Helper Cell Development via Regulating Chromatin Accessibility*“, we want to point out that although the title of this study focusses on TOX2, very similar observations have been made with TOX in this study, albeit via different cytokine pathways. We cite from this manuscript: “*Next, we assessed whether Tox was similarly regulated as Tox2. We found binding peaks of Bcl6 at the Tox locus by analyzing our previous Bcl6 ChIP-seq data [...] (Figure S6C; [...]), so Tox is likely a direct target of Bcl6 in Tfh cells too. To confirm this, we analyzed Tox mRNA expression with the same experimental strategy in Figure S1E and found that Tox expression was suppressed in the absence of Bcl6 in CD4⁺CD44^{hi}CXCR5⁺ T cells (Figure S6D), which supports that Tox expression depends on Bcl6. Meanwhile, enforced Bcl6 expression increased Tox transcription compared with cells infected with empty vector under Th0 conditions (Figure S6E). These data demonstrate a role of Bcl6 in regulation of Tox expression, consistent with the previous microarray results using human CD4⁺ T cells (Figure S6F; [...]). However, Tox expression could not be increased by IL-6 and IL-21 but was enhanced by an IL-2-blocking antibody under neutral condition (Figure S6G). Therefore, Tox2 and Tox are both directly regulated by Bcl6 during Tfh development but differently mediated by IL-6 and IL-21 signaling pathways.*“

222 “Interestingly, BL vaccine-specific antibodies were highest”. This finding is not unexpected. Please consider re-phrasing.

Response: We changed our wording and removed the word “Interestingly”.

Discussion

More thought needs to go into the discussion around the function of cTfh and what these data show that adds to previous work in the field. For example, the frequency of CD38⁺ cTfh was higher post immunisation with influenza vaccine in those who had not been immunised in the

preceding three years compared with those who had (M Cole et al. Responses to Quadrivalent Influenza Vaccine Reveal Distinct Circulating CD4+CXCR5+ T Cell Subsets in Men Living with HIV | Scientific Reports (nature.com))

Response: We very much appreciate the reviewer's comment and have put more details into the discussion and the previously published literature. Regarding the elegant study by Cole et al., we have specifically focused on the different methodological approach we used in our study (Analysis of virus-specific versus bulk CD4 T cell responses and peak of the T cell activation response day 4 versus day 7). More specifically, discussing our data in the context of the work by Cole et al. two main conclusions could be derived. First, although Cole et al. did not analyze influenza-specific CD4 T cells, the increase of CD38 and ICOS expression on bulk cTfh cells is likely a reflection of vaccine-reactive influenza-specific CD4 T cells as we have not observed relevant upregulation of these markers on CD4 T cells that are specific for an influenza epitope that is not included in the vaccine strain. Thus, there appears to be little bystander activation. Second, our observation that only a median of around 40% of CD38 and ICOS expressing HA₁₁₈₋₁₃₂-specific CD4 T cells display a cTfh phenotype (Figure 4d) clearly demonstrates that a large number of vaccine reactive CD4 T cells are not being picked up in studies pre-selecting cTfh cells without identifying their antigen-specificity. The revised parts of the discussion have already been outlined in response to an earlier comment by the reviewer.

295 "In addition, we were able to confirm the Tfh lineage of CXCR5+PD-1+ CD4 T cells on the transcriptional level." This assertion overestimates the impact of the data; please also see comment in the results. Refrasing

Response: We revised our wording. The revised version now reads:

Lines 249f: *The observation that TOX expression is induced after vaccination, most strongly on cells with a cTfh phenotype suggests that TOX might be useful as a transcriptional surrogate of Tfh lineage commitment in cTfh cells (Fig. 4f).*

298 The fact that Bcl-6 is not detectable in cTfh may reflect differences in Tfh and cTfh.

Response: We agree with the reviewer, this is indeed the case. Bcl6 is typically not expressed on circulating Tfh cells, as least not to relevant levels or similar levels compared to lymphoid-tissue derived Tfh cells. We have clarified this with stronger emphasis in the revised version as outlined earlier.

307 There was a difference in phenotype at baseline (CD127 expression). There needs to be a more thorough discussion of the importance and relevance of CD127 in T cell function and long-term maintenance of T cell subsets.

Response: We fully agree with the reviewer's comment that this interesting aspect requires more attention. We have added this argumentation in the revised manuscript and this now reads as follows:

Lines 252ff: However, it remains to be demonstrated which mechanisms prevent CD4 T cell activation in those that receive repeat vaccinations. One possible explanation could be a refractory state of T cells after repeat circles of re-activation, which would be supported by the expression patterns of CD127, the IL-7-receptor alpha-chain. CD127 expression identifies T cells with memory characteristics (61, 62) as signaling via CD127 promotes survival and maintenance of long-lived memory T cells (63). In chronic HCV infection, the ability of HCV-specific CD8 T cells to proliferate is closely associated with the expression of CD127. Indeed, HCV-specific CD8 T cells can be separated into memory-like cells with retained functional capacities that express high levels of CD127 expression and terminally exhausted CD8 T cells with lower CD127 expression (64, 65). Similarly, in our cohort, high CD127 expression at baseline was associated with strong activation 4 days after vaccination. Subsequently, CD127 was downregulated on proliferating cells (66) and CD127 expressing cells dominated the specific CD4 T cell population 28 days after vaccination when the antigen was no longer available (67). Thus, low CD127 expression because of repeat antigenic stimulation might be one factor responsible for the different activation kinetics of HA118-132-specific CD4 T cells. There is discussion of “negative interference” but no consideration of immune regulation. The immune response is necessarily finite, and it may not be an advantage to generate a strong antibody response in the context of already high titres of circulating antibody; it is expected that this response would be regulated.

There is discussion of “negative interference” but no consideration of immune regulation. The immune response is necessarily finite, and it may not be an advantage to generate a strong antibody response in the context of already high titres of circulating antibody; it is expected that this response would be regulated.

Response: This is indeed another interesting comment. We very much agree that it is highly likely that vaccine responses are regulated, although little is known about how this regulation precisely occurs. The theory that “it may not be an advantage to generate a strong antibody response in the context of already high titres of circulating antibody” is certainly valid and needs to be validated, but may also be very context-specific. In contrast to this theory, recent data from individuals with SARS-CoV-2 vaccination after natural infection show stronger antibody responses than those with either natural infection or vaccination alone. Thus, the observations suggest that “repeated exposures are recognized as an increased threat” (32). This would suggest that the host may strive for even stronger antibody responses after repeat exposures even in the context of high pre-existing antibody titers. However, at some level, this will most likely also be regulated by some rheostat. We have therefore added an additional section on immune regulation to the discussion, this now reads:

Lines 281ff: In addition, the inverse correlation between preexisting antibodies and the antibody induction may suggest that a certain target level of antibodies specific for a given antigen is centrally regulated(4, 5, 27). This assumption would be supported by our observation that antibody levels were similar between groups at day 28 (Fig. 5c), suggesting that an immune-rheostat strives for a certain target level and prevents immune activation in response to the vaccine if the target level is already reached. In contrast to this hypothesis, it has recently been shown that individuals receiving SARS-CoV-2 vaccination after natural infection show stronger antibody responses than those with either natural infection or vaccination alone(28-31). The sequence of natural infection followed by vaccination results in a state termed “hybrid immunity” and the higher antibody titers observed in these individuals has been attributed to

the idea that the host perceives repeat antigenic exposure as an increased threat and subsequently strives for higher antibody titers(32).

329-335. This is speculation. Where are the data that mRNA vaccines are more stable inducers of T cell activation than vaccines of other design, particularly adenoviral vectored vaccines?

Response: We agree with the reviewer that this statement has speculative aspects and have therefore removed this from the discussion, although we believe that the biological background for this speculation is valid and that recent upcoming publications demonstrate that mRNA vaccines are indeed strong inducers of T cell responses (29, 68) without being able, however, to compare these responses to those observed in our study with a different pathogen and a different background immunity.

Methods

These need a timeline and clearer description of the groups and vaccination history.

Response: The restructuring of the manuscript with the individual symbolism for each donor/year and the color classification according to vaccination history makes the assignment clearer (Fig 1). In addition, the Table 1 contains all information on the donors.

Tables

Where are the demographics of participants – age, sex, ethnicity etc?

Response: Donor characteristics are included in the Table 1. However we have added the ethnicity as it was not stated yet.

Figures

Fig 2 What is the gating strategy for the t-SNE?

Response: We thank the reviewer for this remark and have added the missing gating strategy for the t-SNE to the Supplementary Fig. 11.

Although divided into three clusters, the data appear more like four clusters. Cluster 2 looks like two clusters based on CXCR5 expression: one that is CXCR5 lo and one that is CXCR5 mid to hi. CXCR5 hi cells in this cluster express high levels of cTfh activation markers. There also appears to be a cluster of CXCR5+ cells that appear in cluster three.

Response: We thank the reviewer for this comment that we had also discussed prior to drafting the manuscript as clustering algorithms can provide different degrees of complexity and resulting clusters. An increase in the number of clusters results in more homogeneous clusters, but increases the complexity of the analyses. We believe that there are good arguments for maintaining three clusters, but there is also an argument to be made for 4 clusters (which results in a separation of cluster 2 as outlined by the reviewer). Thus, we reran the omig

analysis, divided into four clusters. In the resulting figure, HA₁₁₈₋₁₃₂ specific CD4 T cells were labeled according to the vaccination history (Fig3c) and their activation phenotype (early vs. late). Cluster 1 represents cells with a resting memory phenotype (CD127+, TCF-1+), cluster 2 represents strongly activated and proliferating Tfh cells (CD38+, ICOS+, Ki67+, CXCR5, PD-1 among others), cluster 3 displays strongly activated and proliferating cells with less Tfh cell character (CD38+, ICOS+, Ki67+, among others) and cluster 4 includes cells with a moderately activated, effector phenotype (CD127low, CD38low, T-bet high). With this analysis we defined another cluster that separated the acitivated and proliferating cells into cells with stronger cTfh marker expression and those with weaker cTfh-associated marker expression. While these two clusters are now separated, they both remain populated by HA₁₁₈₋₁₃₂ specific CD4 T cells from individuals with an “early” activation pattern that largely overlap with thos individuals that are not recently vaccinated.

Fig 4 Difference in fold change between early and late is driven by four individuals – these findings are overstated in the text. Is there anything different about these individuals that can be detected clinically or immunologically? This affects every panel of this figure and therefore needs addressing.

Response: We appreciate the reviewer’s comment and have addressed this aspect in the revised version of the manuscript. As discussed in response to reviewer #1, we have now assigned the vaccinees to groups based on their vaccination history. With regards to the antibody response against the entire vaccine, shown in the initial submission, the vaccination history is a strong separator and those early responders not showing strong antibody responses against the entire vaccine belong to the recently vaccinated group. However, in response to reviewer #2, we have performed new serologies, specifically analyzing H3-specific antibody responses. These new data largely confirm the results obtained by analyzing antibody responses against the entire vaccine but more closely associate the T cell activation patterns with the antibody response. Collectively, our data now demonstrate that an early activation of HA₁₁₈₋₁₃₂ –specific CD4 T cell reponses is predictive of an increase in H3-specific antibodies after influenza vaccination, more so than the vaccination history alone.

Fig 5 Is the expression of CD127 highest in those driving the changes in Fig 4? Are there the same four individuals in panels e and f as in Fig 4?

Response: This is indeed correct, these are the same donors. By displaying patients with individual symbols, assigned according to their vaccination history, traceability becomes easier (see Supplementary Fig. 8).

Supplementary Tables and Figures

Supplementary Fig 2e It appears there are at least two individuals with ICOS+CD38++ HA 306-318-specific CD4 T cells at Day 7 and beyond; please comment on this in the text. Where are the data demonstrating no change in the total frequency of HA 306-318-specific CD4 T cells rather than the frequency of activated cells?

Response: We agree with the first comment and apologize for creating confusion by not labeling the axes for both T cell specificities with the same range, this has been corrected in the revised manuscript. Indeed, the graph the reviewer is referring to, shows two donors demonstrating an increase to 4% and 8% of HA₃₀₆₋₃₁₈-specific CD4 T cells expressing ICOS and CD38 at day 7. These are significantly lower activation signals compared to CD4 T cells targeting the conserved epitope (HA₁₁₈₋₁₃₂). It might be important to point out, however, that this minimal activation of HA₃₀₆₋₃₁₈- specific CD4 T cells might in fact be due to low avidity activation. Indeed, these two individuals were vaccinated with a strain harboring only 2 amino acid substitutions in the HA₃₀₆₋₃₁₈-epitope, compared to three substitutions in the vaccine strain of the other individuals. While these observations are immunologically highly interesting, the limited data set and the different main focus of our work prevented us from describing this side note in detail. If the reviewer and the editorial would like to see this information in the main manuscript, we are happy to revise accordingly.

Regarding the second part of the question, figure 1d shows the frequency over time of HA 306-318-specific CD4 T cells. The activation markers expressed on these HA306-318-specific CD4 T cells are shown in Supplementary Fig. 2. In both cases, no significant changes were observed over time.

Fig4c Are there baseline data for 2-NBDG uptake?

Response: We have analyzed the longitudinal uptake of 2-NBDG. Therefore we normalized the MFI of 2-NBDG to naive CD4 T cells. We could not detect a significant longitudinal difference of glucose uptake. In order to address this point and make our experimental setup and results clearer, we decided to add graph to the revised Fig 4a.

Minor comments

The words interestingly and importantly are used repeatedly throughout. Please consider re-phrasing these sentences and allow the reader to decide on these aspects. Check for grammatical and spelling errors

Response: We have rewritten large parts of the manuscript and have tried to avoid the words interestingly and importantly as per the reviewer's suggestion. We have also checked for grammatical and spelling errors

References

1. S. E. Bentebibel *et al.*, Induction of ICOS+CXCR3+CXCR5+ TH cells correlates with antibody responses to influenza vaccination. *Sci Transl Med* **5**, 176ra132 (2013).
2. S. E. Bentebibel *et al.*, ICOS+PD-1(+)+CXCR3(+) T follicular helper cells contribute to the generation of high-avidity antibodies following influenza vaccination. *Scientific Reports* **6**, 26494 (2016).
3. R. S. Herati *et al.*, Circulating CXCR5(+)+PD-1(+) Response Predicts Influenza Vaccine Antibody Responses in Young Adults but not Elderly Adults. *Journal of Immunology* **193**, 3528-3537 (2014).

4. M. Koutsakos *et al.*, Circulating T(FH) cells, serological memory, and tissue compartmentalization shape human influenza-specific B cell immunity. *Sci Transl Med* **10**, Feb 14;10(428):eaan8405 (2018).
5. M. E. Cole *et al.*, Responses to Quadrivalent Influenza Vaccine Reveal Distinct Circulating CD4+CXCR5+ T Cell Subsets in Men Living with HIV. *Sci Rep* **9**, 15650 (2019).
6. R. S. Herati *et al.*, Successive annual influenza vaccination induces a recurrent oligoclonotypic memory response in circulating T follicular helper cells. *Sci Immunol* **2**, Feb;2(8):eaag2152 (2017).
7. F. Spensieri *et al.*, Human circulating influenza-CD4(+) ICOS1(+)IL-21(+) T cells expand after vaccination, exert helper function, and predict antibody responses. *Proceedings of the National Academy of Sciences of the United States of America* **110**, 14330-14335 (2013).
8. F. Spensieri *et al.*, Early Rise of Blood T Follicular Helper Cell Subsets and Baseline Immunity as Predictors of Persisting Late Functional Antibody Responses to Vaccination in Humans. *Plos One* **11**, 11(6): e0157066 (2016).
9. K. A. Richards *et al.*, Evidence That Blunted CD4 T-Cell Responses Underlie Deficient Protective Antibody Responses to Influenza Vaccines in Repeatedly Vaccinated Human Subjects. *J Infect Dis* **222**, 273-277 (2020).
10. H. Uchtenhagen *et al.*, Efficient ex vivo analysis of CD4+ T-cell responses using combinatorial HLA class II tetramer staining. *Nat Commun* **7**, 12614 (2016).
11. M. A. Pilkinton *et al.*, Greater activation of peripheral T follicular helper cells following high dose influenza vaccine in older adults forecasts seroconversion. *Vaccine* **35**, 329-336 (2017).
12. K. A. Huang, S. C. Chang, Y. C. Huang, C. H. Chiu, T. Y. Lin, Antibody Responses to Trivalent Inactivated Influenza Vaccine in Health Care Personnel Previously Vaccinated and Vaccinated for The First Time. *Sci Rep* **7**, 40027 (2017).
13. S. Khurana *et al.*, Repeat vaccination reduces antibody affinity maturation across different influenza vaccine platforms in humans. *Nat Commun* **10**, 3338 (2019).
14. W. A. Keitel, T. R. Cate, R. B. Couch, L. L. Huggins, K. R. Hess, Efficacy of repeated annual immunization with inactivated influenza virus vaccines over a five year period. *Vaccine* **15**, 1114-1122 (1997).
15. W. E. Beyer *et al.*, Effects of repeated annual influenza vaccination on vaccine sero-response in young and elderly adults. *Vaccine* **14**, 1331-1339 (1996).
16. T. W. Y. Ng *et al.*, The Effect of Influenza Vaccination History on Changes in Hemagglutination Inhibition Titers After Receipt of the 2015-2016 Influenza Vaccine in Older Adults in Hong Kong. *Journal of Infectious Diseases* **221**, 33-41 (2020).
17. D. M. Skowronski *et al.*, Serial Vaccination and the Antigenic Distance Hypothesis: Effects on Influenza Vaccine Effectiveness During A(H3N2) Epidemics in Canada, 2010-2011 to 2014-2015. *Journal of Infectious Diseases* **215**, 1059-1069 (2017).
18. J. A. Lewnard, S. Cobey, Immune History and Influenza Vaccine Effectiveness. *Vaccines* **6**, 28 (2018).
19. A. Zhang, H. D. Stacey, C. E. Mullarkey, M. S. Miller, Original Antigenic Sin: How First Exposure Shapes Lifelong Anti-Influenza Virus Immune Responses. *The Journal of Immunology* **202**, 335-340 (2019).
20. V. I. Zarnitsyna *et al.*, Masking of antigenic epitopes by antibodies shapes the humoral immune response to influenza. *Philosophical Transactions of the Royal Society B: Biological Sciences* **370**, 20140248 (2015).
21. V. I. Zarnitsyna, J. Lavine, A. Ellebedy, R. Ahmed, R. Antia, Multi-epitope Models Explain How Pre-existing Antibodies Affect the Generation of Broadly Protective Responses to Influenza. *PLOS Pathogens* **12**, e1005692 (2016).
22. A. Lanzavecchia, Mechanisms of antigen uptake for presentation. *Current Opinion in Immunology* **8**, 348-354 (1996).
23. A. O. Kamphorst, P. Guermonprez, D. Dudziak, M. C. Nussenzweig, Route of Antigen Uptake Differentially Impacts Presentation by Dendritic Cells and Activated Monocytes. *The Journal of Immunology* **185**, 3426-3435 (2010).

24. A. Regnault *et al.*, Fcγ Receptor–mediated Induction of Dendritic Cell Maturation and Major Histocompatibility Complex Class I–restricted Antigen Presentation after Immune Complex Internalization. *Journal of Experimental Medicine* **189**, 371-380 (1999).
25. A. Zhang, H. D. Stacey, C. E. Mullarkey, M. S. Miller, Original Antigenic Sin: How First Exposure Shapes Lifelong Anti-Influenza Virus Immune Responses. *Journal of Immunology* **202**, 335-340 (2019).
26. H. Jang, T. M. Ross, Preexisting influenza specific immunity and vaccine effectiveness. *Expert Review of Vaccines* **18**, 1043-1051 (2019).
27. X. S. He *et al.*, Baseline levels of influenza-specific CD4 memory T-cells affect T-cell responses to influenza vaccines. *PLoS One* **3**, e2574 (2008).
28. L. Stamatatos *et al.*, mRNA vaccination boosts cross-variant neutralizing antibodies elicited by SARS-CoV-2 infection. *Science*, 11(6):e015706 (2021).
29. C. J. Reynolds *et al.*, Prior SARS-CoV-2 infection rescues B and T cell responses to variants after first vaccine dose. *Science*, Vol. 372, Issue 6549, pp. 1418-1423 (2021).
30. R. R. Goel *et al.*, Distinct antibody and memory B cell responses in SARS-CoV-2 naïve and recovered individuals following mRNA vaccination. *Sci Immunol* **6**, Vol. 6, Issue 58, eabi6950 (2021).
31. D. Geers *et al.*, SARS-CoV-2 variants of concern partially escape humoral but not T-cell responses in COVID-19 convalescent donors and vaccinees. *Sci Immunol* **6**, Vol. 6, Issue 59, eabj1750 (2021).
32. S. Crotty, Hybrid immunity. *Science* **372**, 1392-1393 (2021).
33. H. Hon, A. Oran, T. Brocker, J. Jacob, B lymphocytes participate in cross-presentation of antigen following gene gun vaccination. *J Immunol* **174**, 5233-5242 (2005).
34. C. M. Fehres, W. W. Unger, J. J. Garcia-Vallejo, Y. van Kooyk, Understanding the biology of antigen cross-presentation for the design of vaccines against cancer. *Front Immunol* **5**, 149 (2014).
35. F. Krammer, L. Li, P. C. Wilson, Emerging from the Shadow of Hemagglutinin: Neuraminidase Is an Important Target for Influenza Vaccination. *Cell Host Microbe* **26**, 712-713 (2019).
36. A. M. Avalos, H. L. Ploegh, Early BCR Events and Antigen Capture, Processing, and Loading on MHC Class II on B Cells. *Front Immunol* **5**, 92 (2014).
37. L. N. Adler *et al.*, The Other Function: Class II-Restricted Antigen Presentation by B Cells. *Front Immunol* **8**, 319 (2017).
38. D. C. Parker, T cell-dependent B cell activation. *Annu Rev Immunol* **11**, 331-360 (1993).
39. A. O. Kamphorst, P. Guermonprez, D. Dudziak, M. C. Nussenzweig, Route of antigen uptake differentially impacts presentation by dendritic cells and activated monocytes. *J Immunol* **185**, 3426-3435 (2010).
40. A. Lanzavecchia, Mechanisms of antigen uptake for presentation. *Curr Opin Immunol* **8**, 348-354 (1996).
41. W. G. Land, in *Damage-Associated Molecular Patterns in Human Diseases*. (Springer, Cham., 2018).
42. F. Junker, J. Gordon, O. Qureshi, Fc Gamma Receptors and Their Role in Antigen Uptake, Presentation, and T Cell Activation. *Front Immunol* **11**, 1393 (2020).
43. M. Guilliams, P. Bruhns, Y. Saeys, H. Hammad, B. N. Lambrecht, The function of Fcγ receptors in dendritic cells and macrophages. *Nat Rev Immunol* **14**, 94-108 (2014).
44. S. Amigorena, J. Salamero, J. Davoust, W. H. Fridman, C. Bonnerot, Tyrosine-containing motif that transduces cell activation signals also determines internalization and antigen presentation via type III receptors for IgG. *Nature* **358**, 337-341 (1992).
45. A. Regnault *et al.*, Fcγ receptor-mediated induction of dendritic cell maturation and major histocompatibility complex class I-restricted antigen presentation after immune complex internalization. *J Exp Med* **189**, 371-380 (1999).

46. W. H. Fridman, Regulation of B-cell activation and antigen presentation by Fc receptors. *Curr Opin Immunol* **5**, 355-360 (1993).
47. J. V. Ravetch, S. Bolland, IgG Fc receptors. *Annu Rev Immunol* **19**, 275-290 (2001).
48. M. Espéli *et al.*, FcγRIIb differentially regulates pre-immune and germinal center B cell tolerance in mouse and human. *Nat Commun* **10**, 1970 (2019).
49. S. Crotty, Follicular helper CD4 T cells (TFH). *Annu Rev Immunol* **29**, 621-663 (2011).
50. L. A. Vella *et al.*, T follicular helper cells in human efferent lymph retain lymphoid characteristics. *J Clin Invest* **129**, 3185-3200 (2019).
51. A. Heit *et al.*, Vaccination establishes clonal relatives of germinal center T cells in the blood of humans. *J Exp Med* **214**, 2139-2152 (2017).
52. M. Locci *et al.*, Human circulating PD-1+CXCR3-CXCR5+ memory Tfh cells are highly functional and correlate with broadly neutralizing HIV antibody responses. *Immunity* **39**, 758-769 (2013).
53. N. Simpson *et al.*, Expansion of circulating T cells resembling follicular helper T cells is a fixed phenotype that identifies a subset of severe systemic lupus erythematosus. *Arthritis Rheum* **62**, 234-244 (2010).
54. C. S. Ma, E. K. Deenick, M. Batten, S. G. Tangye, The origins, function, and regulation of T follicular helper cells. *J Exp Med* **209**, 1241-1253 (2012).
55. A. U. Rasheed, H. P. Rahn, F. Sallusto, M. Lipp, G. Muller, Follicular B helper T cell activity is confined to CXCR5(hi)ICOS(hi) CD4 T cells and is independent of CD57 expression. *Eur J Immunol* **36**, 1892-1903 (2006).
56. S. Crotty, T Follicular Helper Cell Biology: A Decade of Discovery and Diseases. *Immunity* **50**, 1132-1148 (2019).
57. J. S. Hale, R. Ahmed, Memory T follicular helper CD4 T cells. *Front Immunol* **6**, 16 (2015).
58. J. A. Gullicksrud *et al.*, Differential Requirements for Tcf1 Long Isoforms in CD8(+) and CD4(+) T Cell Responses to Acute Viral Infection. *Journal of immunology* **199**, 911-919 (2017).
59. R. S. Herati *et al.*, Successive annual influenza vaccination induces a recurrent oligoclonotypic memory response in circulating T follicular helper cells. *Sci Immunol* **2**, eaag2152 (2017).
60. W. Xu *et al.*, The Transcription Factor Tox2 Drives T Follicular Helper Cell Development via Regulating Chromatin Accessibility. *Immunity* **51**, 826-839 e825 (2019).
61. T. Boettler *et al.*, Expression of the interleukin-7 receptor alpha chain (CD127) on virus-specific CD8+ T cells identifies functionally and phenotypically defined memory T cells during acute resolving hepatitis B virus infection. *J Virol* **80**, 3532-3540 (2006).
62. K. K. McKinstry *et al.*, Effector CD4 T-cell transition to memory requires late cognate interactions that induce autocrine IL-2. *Nat Commun* **5**, 5377 (2014).
63. M. D. Martin, V. P. Badovinac, Defining Memory CD8 T Cell. *Front Immunol* **9**, 2692 (2018).
64. N. Hensel *et al.*, Memory-like HCV-specific CD8(+) T cells retain a molecular scar after cure of chronic HCV infection. *Nat Immunol* **22**, 229-239 (2021).
65. D. Wieland *et al.*, TCF1(+) hepatitis C virus-specific CD8(+) T cells are maintained after cessation of chronic antigen stimulation. *Nat Commun* **8**, 15050 (2017).
66. K. M. Huster *et al.*, Selective expression of IL-7 receptor on memory T cells identifies early CD40L-dependent generation of distinct CD8+ memory T cell subsets. *Proc Natl Acad Sci U S A* **101**, 5610-5615 (2004).
67. M. Smits *et al.*, Follicular T helper cells shape the HCV-specific CD4+ T cell repertoire after virus elimination. *J Clin Invest* **130**, 998-1009 (2020).
68. B. A. Woldemeskel, C. C. Garliss, J. N. Blankson, SARS-CoV-2 mRNA vaccines induce broad CD4+ T cell responses that recognize SARS-CoV-2 variants and HCoV-NL63. *J Clin Invest* **131**, e149335 (2021).

REVIEWERS' COMMENTS

Reviewer #1 (Remarks to the Author):

The authors have made a comprehensive response to the issues raised in the review and more ability to track individual responses is helpful as is the more scholarly treatment of the literature on this interesting topic. The detail in the analyses of the tetramer specific cells and the sampling at different time points post vaccination is very comprehensive and valuable adding considerable depth and novelty to this study.

Despite this, the manuscript is still extremely dense and most of the differences expressed by the CD4 T cells are quite modest. Also, the key differences now available show that subjects with no vaccine history are distinct, more robust and generally, the expansion of relevant CD4 T cells tracked with a tetramer occur earlier than those with vaccine history. However, as before, there is considerable (though not surprisingly) heterogeneity in the kinetics of the response and expression of markers, and the classification of "early" vs "late" is still quite confusing because of this and this "lumping" of responders does not seem useful. Clearly there are early responders in the 3/4 of the unvaccinated group), which is interesting. Because this study tracks only one specificity, the CD4 T cells specific for this highly conserved epitope may have a history of repeated stimulation, as will some of the B cells. So the behavior of these CD4 T cells may be atypical. Also, the cells that emerge at this early time point may be those that have participated in the extrafollicular rather than germinal center response, which apparently lasts for many weeks post vaccination. This is not addressed by the authors. The pattern of more robust responses in individuals who were not vaccinated in the previous season has now been documented by number of groups, which the authors now cite, but this point has been made in the literature.

However, this is still only 1 single epitope in all of H3, and the sample size is extremely small. Therefore, the global conclusions are not warranted. The diversity of CD4 T cells in humans that vary in immune history (infection vs vaccination) and number of times restimulated, as well as the role of relatively "newly elicited" CD4 T cells to drift variants could lead to epitope specific patterns.

The title and abstract need to specify that this study is "tracking a single specificity in detail" over time. The title and abstract of this this revised manuscript is far too extreme and not warranted due to the limitations of the study. T

I would recommend extracting the key novel points, the kinetics and markers that are tracked that reveal the most compelling conclusions. This would allow the reader to quickly grasp what has been done in this study.

Reviewer #2 (Remarks to the Author):

The authors adequately addressed all comments raised during the previous round of reviews.

Reviewer #3 (Remarks to the Author):

The authors present an updated manuscript with re-analysis of initial findings and new data from HA serology (HA recombinant-specific IgG). The original manuscript was subject to a detailed review from three reviewers, to which the authors have clearly responded in all sections of the work. The presentation and description of findings is now clear. The writing style is much improved with better placement of the relevance of these findings within the current literature. Taken together, this manuscript is a well presented and interesting work in its own right and will stimulate further research in this important area. I recommend for publication.

Point by Point letter

Response

We sincerely thank the reviewers for spending their precious time with our manuscript and for providing their valuable comments.

Reviewer #1 (Remarks to the Author):

The authors have made a comprehensive response to the issues raised in the review and more ability to track individual responses is helpful as is the more scholarly treatment of the literature on this interesting topic. The detail in the analyses of the tetramer specific cells and the sampling at different time points post vaccination is very comprehensive and valuable adding considerable depth and novelty to this study.

Despite this, the manuscript is still extremely dense and most of the differences expressed by the CD4 T cells are quite modest. Also, the key differences now available show that subjects with no vaccine history are distinct, more robust and generally, the expansion of relevant CD4 T cells tracked with a tetramer occur earlier than those with vaccine history. However, as before, there is considerable (though not surprisingly) heterogeneity in the kinetics of the response and expression of markers, and the classification of "early" vs "late" is still quite confusing because of this and this "lumping" of responders does not seem useful. Clearly there are early responders in the 3/4 of the unvaccinated group), which is interesting. Because this study tracks only one specificity, the CD4 T cells specific for this highly conserved epitope may have a history of repeated stimulation, as will some of the B cells. So the behavior of these CD4 T cells may be atypical. Also, the cells that emerge at this early time point may be those that have participated in the extrafollicular rather than germinal center response, which apparently lasts for many weeks post vaccination. This is not addressed by the authors. The pattern of more robust responses in individuals who were not vaccinated in the previous season has now been documented by number of groups, which the authors now cite, but this point has been made in the literature.

However, this is still only 1 single epitope in all of H3, and the sample size is extremely small. Therefore, the global conclusions are not warranted. The diversity of CD4 T cells in humans that vary in immune history (infection vs vaccination) and number of times restimulated, as well as the role of relatively "newly elicited" CD4 T cells to drift variants could lead to epitope specific patterns.

The title and abstract need to specify that this study is "tracking a single specificity in detail" over time. The title and abstract of this this revised manuscript is far too extreme and not warranted due to the limitations of the study.

I would recommend extracting the key novel points, the kinetics and markers that are tracked that reveal the most compelling conclusions. This would allow the reader to quickly grasp what has been done in this study.

Response:

We thank the reviewer for the positive evaluation of our study, for highlighting the methodological strengths as well as the changes regarding the traceability of the individual donors. We also appreciate the remaining concern of reviewer#1. Indeed, we can only describe a single epitope of HA and cannot assume that all HA-specific CD4 T cell specificities would behave similarly. This limitation is now reflected in the revised title and abstract. In addition, we have addressed the relevance of the vaccination history in both the title and abstract.

Reviewer #2 (Remarks to the Author):

The authors adequately addressed all comments raised during the previous round of reviews.

Reviewer #3 (Remarks to the Author):

The authors present an updated manuscript with re-analysis of initial findings and new data from HA serology (HA recombinant-specific IgG). The original manuscript was subject to a detailed review from three reviewers, to which the authors have clearly responded in all sections of the work. The presentation and description of findings is now clear. The writing style is much improved with better placement of the relevance of these findings within the current literature. Taken together, this manuscript is a well presented and interesting work in its own right and will stimulate further research in this important area. I recommend for publication.

Response to Reviewer #2 + #3:

We thank the reviewers for the positive assessment of our efforts to address their helpful comments in the first round of revisions.